

# Non-destructive estimates of soil carbonic anhydrase activity and soil water oxygen isotope composition

Sam P. Jones[1], Jérôme Ogée[1], Joana Sauze[1], Steven Wohl[1], Noelia Saavedra[1], Noelia Fernández-Prado[1], Juliette Maire[1], Thomas Launois[1], Alexandre Bosc[1], and Lisa Wingate[1]

[1] ISPA, Bordeaux Science Agro, INRA, Villenave d'Ornon 33140, France.

*Correspondence to*: Sam Jones (samuel.jones@inra.fr)

**Abstract**

The contribution of photosynthesis and soil respiration to net land-atmosphere carbon dioxide ($CO_2$) exchange can be
estimated based on the differential influence of leaves and soils on budgets of the oxygen isotope composition ($\delta^{18}O$) of
atmospheric $CO_2$. To do so, the activity of carbonic anhydrases (CA), a group of enzymes that catalyse the hydration of $CO_2$,
in soils and plants needs to be understood. Measurements of soil CA activity typically involve the inversion of models
describing the $\delta^{18}O$ of $CO_2$ fluxes to solve for the apparent, potentially catalysed, rate of $CO_2$ hydration. This requires
information about the $\delta^{18}O$ of $CO_2$ in isotopic equilibrium with soil water, typically obtained from destructive, depth-
resolved sampling and extraction of soil water. In doing so, an assumption is made about the soil water pool that $CO_2$
interacts with, that may bias estimates of CA activity if incorrect. Furthermore, this can represent a significant challenge in
data collection given the potential for spatial and temporal variability in the $\delta^{18}O$ of soil water and limited *a priori*
information with respect to the appropriate sampling resolution and depth. We investigated whether we could circumvent this
requirement by inferring the rate of $CO_2$ hydration and the $\delta^{18}O$ of soil water from the relationship between the $\delta^{18}O$ of $CO_2$
fluxes and the $\delta^{18}O$ of $CO_2$ at the soil surface measured at different ambient $CO_2$ conditions. This approach was tested
through laboratory incubations of air-dried soils that were re-wetted with three waters of different $\delta^{18}O$. Gas exchange
measurements were made on these soils to estimate the rate of hydration and the $\delta^{18}O$ of soil water, followed by soil water
extraction to allow for comparison. Estimated rates of $CO_2$ hydration were 6.8 to 14.6 times greater than the theoretical un-
catalysed rate of hydration, indicating that CA were active in these soils. Importantly, these estimates were not significantly
different among water treatments suggesting that this represents a robust approach to assay the activity of CA in soil. As
expected, estimates of the $\delta^{18}O$ of the soil water that equilibrates with $CO_2$ varied in response to alteration to the $\delta^{18}O$ of soil
water. However, these estimates were consistently more negative than the composition of the soil water extracted by
cryogenic vacuum distillation at the end of the gas measurements with differences of up to -3.94 ‰ VSMOW. These offsets
suggest that $CO_2$ may be principally interacting with water pools associated with particle surfaces rather than the bulk water
pool under the incubation conditions of this study.



# 1 Introduction

Carbonic anhydrases (CA) are a group of metalloenzymes, typically utilising either zinc (Hewett-Emmett and Tashian, 1996) or cadmium (Xu et al., 2008), that catalyse the reversible hydration of dissolved carbon dioxide ($CO_2$). Spread amongst at least five unrelated classes, these enzymes have been identified in eukarya, bacteria and archaea (Gilmour, 2010). Such

convergent evolution among diverse groups of organisms suggests that CA are fundamental to many life strategies (Smith et al., 1999). Indeed, these enzymes have been linked to a number of common and specialised biological processes, such as $CO_2$ concentration mechanisms required to maintain photosynthesis in plants, algae and cyanobacteria (Badger, 2003; Badger and Price, 1994), calcification to limit calcium toxicity in bacteria (Banks et al., 2010; Li et al., 2005b), maintenance of required $CO_2$ and bicarbonate levels for metabolic activity in both bacteria (Merlin et al., 2003) and fungi (Kaur et al.,

2009) growing under $CO_2$ limited conditions, and metabolic flexibility in methanogenic archaea (Smith and Ferry, 2000). However, despite evidence of CA activity in soils, the variability and drivers of their expression by soil communities is poorly understood (Li et al., 2005a; Seibt et al., 2006; Wingate et al., 2009, 2008).

This knowledge gap is of particular importance as soil CA activity can considerably alter the oxygen isotope composition

($\delta^{18}O$) of atmospheric $CO_2$ ($\delta_{atm}$) (Stern et al., 2001; Tans, 1998; Wingate et al., 2009). The presence of CA in soils and leaves influences $\delta_{atm}$ as oxygen isotopes are exchanged between $CO_2$ and water during CA-catalysed hydration (Mills and Urey, 1940; Uchikawa and Zeebe, 2012). The $\delta^{18}O$ of soil ($\delta_{sw}$) and leaf water pools are typically distinct because of differences in pool sizes and evaporation rates, and these different signals are transferred to $CO_2$ molecules interacting with these water pools (Farquhar et al., 1993; Francey and Tans, 1987; Stern et al., 2001). This leads to contrasted $\delta^{18}O$ signatures of soil-

atmosphere ($\delta_R$) and leaf-atmosphere $CO_2$ exchange that can be used to partition the contribution of photosynthesis and soil respiration, the largest gross fluxes in the contemporary atmospheric carbon cycle (Ciais et al., 2013), to the net atmospheric $CO_2$ budget (Welp et al., 2011; Yakir and Wang, 1996). Whilst the extent to which the $\delta^{18}O$ of $CO_2$ interacting with leaves approaches equilibrium with leaf water pools has been considered (Gillon and Yakir, 2001), the degree to which the catalysis of $CO_2$ hydration by CA in soils influences $\delta_R$ is less well understood (Wingate et al., 2009). As such, appropriately

modelling $\delta_{atm}$ budgets relies on improving our knowledge of soil CA activity. In this respect, a better understanding of soil CA activity not only represents a frontier in soil ecology but also in understanding interactions between soil hydrological and carbon cycles, and ecosystem function within the carbon cycle at much larger scales.

A number of methods have been developed to estimate CA activity. Conventionally, assays have expressed activity by

comparing the time required to achieve a set pH change in a $CO_2$-saturated buffer solution in the presence and absence of



CA-containing extracts (Wilbur and Anderson, 1948). Whilst this approach has been applied to soils (Li et al., 2005a), the requirement to work with enzyme extractions at low temperatures implies that activity is being estimated under extremely disturbed conditions. Less disruptive isotope labelling techniques, that estimate the rate of hydration and thus CA activity based on the loss of the label from a $\delta^{18}O$-enriched $CO_2$ source (Mills and Urey, 1940; Tu et al., 1978) have also been applied to studies of aquatic algae (Hopkinson et al., 2013). Similarly, soil studies have focused on inverting models that describe $\delta_R$ (Miller et al., 1999; Tans, 1998) to assay CA activity under realistic conditions from natural abundance gas flux measurements (Kapiluto et al., 2007; Seibt et al., 2006; Wingate et al., 2008). Following Tans (1998) for a semi-infinite soil column with a constant $CO_2$ production profile, $\delta_R$ (‰ $VPDB_g$) at steady state can be described as (see also Wingate et al., 2010):

$$\delta_R = -\frac{v_{inv}C_a}{F_R}\delta_a + \frac{v_{inv}C_a}{F_R}\delta_{eq} + \delta_{eq} - a \quad ,$$

(1)

where $\delta_{eq}$ (‰ $VPDB_g$) is the $\delta^{18}O$ of $CO_2$ in isotopic equilibrium with $\delta_{sw}$, $a$ (8.8 ‰) is the isotopic fractionation associated with the diffusion of $^{12}C^{16}O^{18}O$ in still air, $v_{inv}$ (m s$^{-1}$) is the so-called piston velocity of $CO_2$ diffusing in and out of the soil, $C_a$ (μmol m$^{-3}$) is the concentration of $CO_2$ in the air at the soil-air interface, $F_R$ (μmol m$^{-2}$ s$^{-1}$) is the net soil-atmosphere $CO_2$ flux and $\delta_a$ (‰ $VPDB_g$) is the $\delta^{18}O$ of $CO_2$ at the soil-air interface. The rate of $^{18}O$ exchange between $CO_2$ and soil water $k_{iso}$ (s$^{-1}$) can be deduced from $v_{inv}$ as:

$$k_{iso} = \frac{v_{inv}^2}{B\theta_w\kappa\phi_a D} \quad ,$$

(2)

where $B$ (m$^3$ m$^{-3}$) is the solubility coefficient for $CO_2$ in water, $\theta_w$ (m$^3$ m$^{-3}$) is the volumetric soil water content, $\kappa$ is the tortuosity of the soil pore network, $\phi_a$ is the soil air-filled porosity and $D$ (m$^2$ s$^{-1}$) is the molecular diffusivity of $^{12}C^{16}O^{18}O$ in air. CA activity is then estimated from $k_{iso}$ from rate constants (Mills and Urey, 1940; Uchikawa and Zeebe, 2012). Either as the equivalent CA concentration required to achieve the observed $k_{iso}$ assuming known enzymatic parameters (Ogée et al., 2016; Uchikawa and Zeebe, 2012) or as the unit-less enhancement factor between $k_{iso}$ and the theoretical un-catalysed rate of $CO_2$-$H_2O$ isotopic exchange $k_{iso,uncat}$ (Seibt et al., 2006; Wingate et al., 2009, 2008). In the first instance, deducing a CA concentration is hampered by the paucity of information regarding the variability of CA kinetic parameters at the bulk soil scale (Ogée et al., 2016) as communities of fungi, algae, bacteria and archaea are likely to express a mixture of α, β and γ-CA classes (Gilmour, 2010; Smith and Ferry, 2000). The second approach has been used to describe temporal variations in CA activity (Seibt et al., 2006; Wingate et al., 2010, 2008), but its meaning is not always intuitive when applied across soil types, given the pH dependency of $k_{iso,uncat}$ (Uchikawa and Zeebe, 2012). This aside, solving Eq. 1 also requires $\delta_{eq}$ to be determined from depth-resolved knowledge about $\delta_{sw}$ (Wingate et al., 2009). In practice, $\delta_{sw}$ has been assumed to be



relatively constant over short periods and either extrapolated from proximal sampling (Seibt et al., 2006; Wingate et al., 2009, 2008) or set by irrigating dried soils with water of known isotopic composition (Kapiluto et al., 2007). Over longer periods, the development of $\delta_{sw}$ vertical profiles has also been estimated using soil water isotope transport models forced by meteorological data (Wingate et al., 2010). Given the need to make an assumption about the soil water pool with which $CO_2$

is interacting, the potential for spatial and temporal variability of $\delta_{sw}$, and limited *a priori* information with respect to appropriate sampling resolution and depth (Miller et al., 1999; Riley, 2005), approaches allowing CA activity to be estimated in the absence of this information are desirable.

Here we explore the idea that solutions for $v_{inv}$ and $\delta_{eq}$ can be obtained as a function of the response of $\delta_R$ to variations in $\delta_a$,

from gas flux measurements. Equation 1 describes a linear relationship of the form $\delta_R = m\delta_a + c$, where the slope, $m$, is $-v_{inv}C_a/F_R$ and the intercept, $c$, is $v_{inv}C_a/F_R\, \delta_{eq} + \delta_{eq} - a$. If $C_a$ and $F_R$ are held constant, $v_{inv}$ and $\delta_{eq}$ can be estimated from a linear regression between $\delta_R$ and $\delta_a$:

$$v_{inv} = \frac{-m}{C_a/F_R},$$

(3a)

$$\delta_{eq} = \frac{c+a}{1-m}$$

(3b)

To test this approach, we conducted laboratory incubations using air-dried soils that were irrigated with three different waters of known $\delta^{18}O$ composition ($\delta_{iw,low}$, $\delta_{iw,med}$ or $\delta_{iw,high}$). The gas fluxes from these incubations were made under three different inlet conditions that varied in terms of their $\delta^{18}O$ composition of $CO_2$ ($\delta_{b,low}$, $\delta_{b,med}$ and $\delta_{b,high}$) but not in terms of total $CO_2$ concentration ($C_b$). Following gas measurements, water was cryogenically extracted from the incubated soils and its isotopic

composition determined ($\delta_{sw,ce}$) to allow for comparison with that estimated from the gas flux measurements ($\delta_{sw,eq}$). We specifically aimed to 1) confirm the suitability of this approach by testing whether $\delta_R$ and $\delta_a$ are linearly related in an experimental context, 2) compare estimates of $\delta_{sw,eq}$ determined from the gas flux measurements with $\delta_{sw,ce}$ measured for the extracted bulk soil water, and 3) compare the sensitivity of $k_{iso}$ estimates to variations in $\delta_{sw}$.

## 2 Materials and methods

### 2.1 Soil sampling and incubation preparation

Soil was collected in April, 2016 from Le Bray, a *Pinus pinaster* plantation in the southwest of France with predominately sandy soil (94.7 % sand, 2.6 % silt and 2.7 % clay), 5% organic carbon and a water-holding capacity of 0.27 g g⁻¹ (Ogée et



al., 2004; Wingate et al., 2010). After the removal of the understory and the litter layer, consisting of living and dead *Molinia caerulea* tussocks, pine needles, cones, and wood fragments, about 6 kg of soil was collected from the superficial 10 cm at 4 locations spaced 5 m apart. This material was mixed and passed through a 4 mm sieve to remove any large debris. Three sub-samples of the sieved soil were taken to determine the pH, water content and $\delta_{sw,ce}$ of the fresh soil at the time of sampling.

The soil was air-dried for approximately two weeks before being stored in a closed box containing desiccant until the incubations were conducted between mid-June and mid-August, 2016. Approximately 400 g of soil were used to determine the $\delta_{sw,ce}$ of any residual soil water and 6 sub-samples were taken to determine pH after drying.

A total of 18 incubations were conducted with 6 replicates each for the $\delta_{iw,low}$, $\delta_{iw,med}$ and $\delta_{iw,high}$ water treatments. Each

incubation was prepared by placing *ca*. 430 g of air-dried soil in a plastic zip lock bag. Approximately 30 g of soil were removed to determine the residual water content and the remaining 400 g were re-wetted inside the bag with 40 ml of $\delta_{iw,low}$, $\delta_{iw,med}$ or $\delta_{iw,high}$ irrigation water. The bag was closed, gently mixed by hand and 300 g of wet soil was then placed into a threaded PTFE chamber with a height of 11.6 cm and an internal diameter of 7.3 cm. The chamber was closed with a screw-top PTFE lid and shaken at 200 rpm for 10 minutes on an orbital shaker. The remaining 100 g of wet soil in the bag was then

sampled for determination of the re-wetted water content and the initial $\delta_{sw,ce}$. Following shaking, the chamber was opened, sharply tapped to encourage the soil to settle in a uniform manner and placed inside a humidifier designed to maintain the soil water content and composition prior to the gas exchange measurements. This consisted of a desiccator filled with 500 ml of the irrigation water through which ambient air was bubbled using a membrane pump. The humidifier was wrapped in reflective foil and kept in a temperature controlled room at 21 °C. Relative humidity and temperature inside the humidifier

was monitored using a small combined sensor and data-logger (Hydrochron, iButtonLink, LLC., USA).

Typically, preparation and pre-incubation of the soils for each water treatment spanned 3 days. Three chambers, staggered at 2 hour intervals, were prepared on each of the first two days. The water used to re-wet the soil and initially fill the humidifier was sampled at the start of each day to characterise its isotopic composition ($\delta_{iw}$). The water in the humidifier was sampled

once on day 2 and twice on day 3 to track any change in its isotopic composition over the course of the pre-incubation. Each chamber was pre-incubated for 24 hours and then removed from the humidifier, weighed to determine water loss and finally connected to the gas exchange system. Following gas exchange measurements, the chamber was immediately removed, and re-weighed to determine water lost during the gas exchange measurements. The depth of the soil ($z_{max}$) inside the chamber was then measured at 4 points using a digital calliper, and 1cm-thick horizons, from 0 to 5 cm depth, removed and split for

determination of depth-resolved water content and $\delta_{sw,ce}$.





Gravimetric water content (GWC) was determined from weight change after oven drying at 105 °C for 24 hours. Soil pH was determined from a soil-to-water slurry of 1:5 and measured in the supernatant after 2 hours. Soil bulk density was calculated from the initial GWC after re-wetting, the wet weight of the soil in the chamber and the volume of the soil in the chamber. Total porosity ($\phi_t$) was calculated from bulk density assuming a particle density of 2.65 g cm$^{-3}$. Volumetric water content ($\theta_w$) was calculated as the product of GWC and bulk density and $\phi_a$ as the difference between $\phi_t$ and $\theta_w$.

## 2.2 Soil water extraction and analysis

Soil samples taken for determination of $\delta_{sw,ce}$ consisted of 20 to 25 g of material stored at 4 °C in 20 ml glass vials with positive insert screw-top caps. Water samples were extracted from these soils using a cryogenic vacuum distillation system based on the design and methodology described by Orlowski et al. (2013). In brief, the system consists of 6 branches each equipped with 4 extraction vessels, a pirani vacuum gauge (APG100-XLC, Edwards, UK) and an isolation valve. The branches connect to a manifold equipped with a vacuum gauge, a vacuum release valve and a two-stage rotary vane vacuum pump (RV5, Edwards, UK). Extractions were prepared by placing about 20 g of soil, topped with oven-baked glass wool, into an extraction vessel. For each branch, extraction vessels were weighed and frozen in a bath of liquid nitrogen. After freezing, the manifold and branches were checked for leaks using the vacuum gauges and evacuated to a starting pressure of less than 0.3 Pa. Extractions were initiated by isolating a branch, transferring the liquid nitrogen bath to a U-shaped water trap and the sample vessels to a water bath initially at room temperature. Extractions lasted for 180 minutes and the water bath was set to 80 °C after 60 minutes. Following extraction, the water traps were removed, the ends sealed with parafilm and the collected ice allowed to thaw before being weighed. The extracted waters were transferred to 5 ml glass vials with positive insert screw-top caps and stored at 4 °C. Extraction vessels and empty water traps were oven-dried and re-weighed to determine extraction efficiency.

The accuracy of cryogenic vacuum distillation techniques has been questioned as the $\delta^{18}$O values of extracted water tends to be depleted, the extent of which depending on soil properties, relative to the irrigation water when oven dried soils that have been re-wetted are considered (Orlowski et al., 2016a; Sprenger et al., 2015). To quantify biases associated with our methodology and the soil studied, two tests were conducted where six 20 g soil samples were re-wetted to a gravimetric water content of about 0.1 g g$^{-1}$ with water that had a $\delta^{18}$O of -4.84 ± 0.06 ‰ VSMOW-SLAP. In the first test, residual water was removed from air-dried soil by oven drying at 105 °C for 24 hours prior to re-wetting. In the second test, residual water was removed by cryogenic extraction following the above methodology prior to re-wetting. These soils were then treated as described for samples.





The $\delta^{18}O$ composition of irrigation and cryogenically-extracted water samples was measured on an off-axis integrated cavity optical spectrometer (TIWA-45EP, Los Gatos Research, USA) coupled to an auto-sampler (Berman et al., 2013). Prior to analysis, 1 ml of each water sample was pipetted into a 1.5 ml vial and capped with a pre-slit septa. An auto-sampler equipped with a 5 μl syringe and a heated septum port (PAL LC-xt, CTC Analytics AG, Switzerland) sequentially injected 8

aliquots of each sample into the analyser. The first three injections for each sample were discarded to avoid inter-sample memory effects (Lis et al., 2008), as were any injections flagged by the analyser's software (Berman et al., 2013). To avoid applying a linearity correction, the mean water density for a run was calculated and any injection with a water density more than $10^{15}$ molecules $cm^{-3}$ away from the mean was rejected (Lis et al., 2008). The mean water density was then re-calculated and only injections within three standard deviations were retained. A raw mean $\delta^{18}O$ was calculated for all samples where 3

to 5 injections were retained. We accounted for analyser drift and report on the VSMOW-SLAP scale by including two working standards (-10.31 ‰ and 0.62 ‰ VSMOW-SLAP) and a quality control (-4.84 ± 0.06 ‰ VSMOW-SLAP) between every 5 samples.

## 2.3 Incubation system and gas analysis

Measurement of gas fluxes at $\delta_{b,low}$, $\delta_{b,med}$ and $\delta_{b,high}$ was achieved using a gas supply manifold capable of delivering one of

three gas mixtures to the inlet of the incubation system (Fig. 1). Briefly, the inlet to the incubation system was connected to a normally closed 2/2 solenoid valve actuated by a microcontroller (Arduino Uno, Arduino LLC, Italy). One port of the valve was connected directly to a cylinder of compressed air ($\delta_{b,high}$) with a total $CO_2$ concentration and $\delta^{18}O$ composition of 424.93 ± 0.01 ppm and -3.45 ± 0.02 ‰ $VPDB_g$ (NOAA Earth System Research Laboratory, USA). The other ports were preceded by open splits and connected to continuous flows from two in-house gas dilution systems capable of providing a

given concentration of $CO_2$ by mixing pure $CO_2$ from gas cylinders into $CO_2$-free air generated by an air compressor (FM2 Atlas Copto, Nacka, Sweden) equipped with a scrubbing column (Ecodry K-MT6, Parker Hannifin, USA). The concentrations of the two gas mixes were adjusted to match that of $\delta_{b,high}$ and contrasts in $\delta^{18}O$ composition were achieved by using pure $CO_2$ cylinders of different origins ($\delta_{b,med}$ and $\delta_{b,low}$). Following the inlet of the incubation system, the gas stream was split into a chamber and bypass line that terminated at open splits in front of a normally closed 2/2 solenoid valve

connected to the sample inlet of a $CO_2$ isotope ratio infra-red spectrometer (Delta Ray IRIS, Thermo Fischer Scientific, Germany). The flow rate of each line was limited to 216 ml $min^{-1}$ using two mass-flow controllers. The soil chamber was connected in-line at the middle of the chamber line and placed in a water bath at 21 °C.

Gas exchange measurements were made by alternately switching the valve to the sample inlet of the IRIS, allowing

approximately 80 ml $min^{-1}$ of the total flow to pass through the instrument, between bypass and chamber lines three times. The IRIS reports concentrations at 1 Hz for the three most abundant isotopologues of $CO_2$ ($^{12}C^{16}O^{16}O$, $^{13}C^{16}O^{16}O$ and



$^{12}C^{18}O^{16}O$), allowing determination of the total concentration and isotopic composition of $CO_2$ (Geldern et al., 2014; Rizzo et al., 2014). The stability of measurements of the total concentration and the $\delta^{18}O$ of $CO_2$ were assessed prior to the gas exchange measurements by analysing the contents of an air cylinder for 22.5 h (Fig. S1) and subsequent computation of Allan variances (Carrio, 2015). A maximum precision of 0.01016 ppm for total $CO_2$ and 0.0487 ‰ $VPDB_g$ for $\delta^{18}O$ was achieved after integrating over 172 s and 144 s, respectively. Given the profile of the Allan plot (Werle, 2010) and the 35 s residence time of the air in the instrument cell (Geldern et al., 2014), a measurement period of 120 s was used. The first 80 s of each measurement were discarded to minimise carry-over effects and the final 40 s were kept and averaged to provide a measurement mean and standard deviation. A 40 s integration period corresponds to standard deviations for total $CO_2$ and the $\delta^{18}O$ of $CO_2$ of 0.0155 ppm and 0.0587 ‰ $VPDB_g$, respectively. Averaged isotopologue concentrations were corrected by bracketing every three pairs of bypass and chamber line measurements with the measurement of two calibration cylinders (Deuste Steinger GmbH, Germany). The standard deviations for total $CO_2$ and the $\delta^{18}O$ of $CO_2$ over 960 s (the interval between two calibration measurements) were 0.0150 ppm and 0.0645 ‰ $VPDB_g$, respectively.

The calibration cylinders contained mixtures of $CO_2$ in synthetic air (21% $O_2$ and 0.93% Ar in a $N_2$ balance) that had been characterised for the total concentration, carbon isotope composition and $\delta^{18}O$ of $CO_2$ (Max Planck Institute for Biogeochemistry IsoLab, Germany). The total concentration, carbon isotope composition and $\delta^{18}O$ of $CO_2$ for each cylinder were, respectively, 380.26 ppm, -3.064 ‰ VPDB and -14.631 ‰ $VPDB_g$, and 481.62 ppm, -3.071 ‰ VPDB and -14.698 ‰ $VPDB_g$. Concentrations of $^{12}C^{16}O^{16}O$, $^{13}C^{16}O^{16}O$ and $^{12}C^{18}O^{16}O$ were calculated from these values following Wen et al. (2013). Measured averages for the calibration cylinders were linearly interpolated and correction coefficients calculated from two-point linear regressions between cylinders, allowing correction of sample isotopologue concentrations and calculation of total concentration and $\delta^{18}O$ of $CO_2$ (Bowling et al., 2003; Wen et al., 2013; Wingate et al., 2010). This calibration scheme was validated for our isotopic measurements by allowing 40 l of pure $CO_2$ to equilibrate with 4 l of water in a 25 l barrel at 24 °C. Approximately three months after filling, the $\delta^{18}O$ of $CO_2$, calibrated using the scheme described above, in the barrel head-space was measured in dilutions with concentrations ranging from 390 to 560 ppm. Following these measurements the water with which the $CO_2$ had equilibrated with was sampled and its $\delta^{18}O$ composition determined as described previously. Measurements of $\delta^{18}O$ of $CO_2$ were not significantly dependent on $CO_2$ concentration over this range but the composition of the $\delta^{18}O$ of $CO_2$ was, on average, 0.31 ‰ more depleted than that of the water. To account for this offset between our gas ($VPDB_g$) and water (VSMOW-SLAP) scales, a post-calibration correction of +0.31 ‰ was applied to all bypass and chamber $CO_2$ measurements.

The soil $CO_2$ flux $F_R$ (µmol m$^{-2}$ s$^{-1}$) was calculated from the corrected values for each pair of calibrated bypass and chamber measurements as:



$$F_R = \frac{u}{A}(C_a - C_b),$$

(5)

where $u$ (mol s$^{-1}$) is the flow rate of dry air through the chamber line, $C_a$ (ppm) is the $CO_2$ concentration of the chamber line, $C_b$ (ppm) is the $CO_2$ concentration of the bypass line and $A$ (m$^2$) is the surface area of the soil in the chamber. Similarly, $\delta_R$ (‰ VPDB$_g$) was calculated as:

$$\delta_R = \frac{\delta_a C_{a,12} - \delta_b C_{b,12}}{C_{a,12} - C_{b,12}},$$

(6)

where $\delta_a$ and $C_{a,12}$ are respectively the $\delta^{18}O$ of $CO_2$ and concentration of $^{12}C^{16}O^{16}O$ in the chamber line and $\delta_b$ and $C_{b,12}$ are the $\delta^{18}O$ of $CO_2$ and concentration of $^{12}C^{16}O^{16}O$ in the bypass line. To ensure steady-state conditions for measurements under different $\delta_b$, a complete cycle where only the chamber line was measured was included to allow the chamber head-space and soil pore-space conditions to stabilise between transitions from atmospheric conditions at the start of the incubation and subsequent $\delta_b$ conditions (Fig. 2). The order of the $\delta_b$ conditions ($\delta_{b,low}$, $\delta_{b,med}$ and $\delta_{b,high}$) also varied between the 6 replicates of each $\delta_{iw}$ treatment to avoid introducing a temporal treatment bias.

## 2.4 Data processing

All data processing and analysis was conducted in R (R Core Team, 2017). Linear regressions of $\delta_R$ and $\delta_a$ were calculated from the measurements taken at the three different combinations of $\delta_b$ for each of the incubations (Bates et al., 2015). Following Eq. 3a, $v_{inv}$ was estimated using the slope $m$ of the linear regression and the mean of the ratio $C_a/F_R$ for each incubation. As Eq. 3 is only strictly valid when there is a semi-infinite soil column (Tans, 1998), we adapted the equation to account for the influence of boundary conditions found at the bottom of the incubation vessel. The soil-depth adjusted piston velocity, $\tilde{v}_{inv}$, was estimated by using $v_{inv}$ obtained from Eq. 3a to iteratively satisfy:

$$v_{inv} = \tilde{v}_{inv} \tanh\left(\frac{\tilde{v}_{inv} z_{max}}{\kappa \phi_a D}\right),$$

(7)

where $z_{max}$ (m) is the total soil column depth. In doing so we adopted the formulation of Moldrup et al. (2003) for the $\kappa$ of repacked soils:

$$\kappa = \frac{\phi_a^{1.5}}{\phi_t},$$

(8)

The temperature-dependant $CO_2$ solubility $B$ was calculated using Weiss (1974) and the diffusivity of $^{12}C^{16}O^{18}O$ in air was calculated according to Massman (1998) and Tans (1998):


$$D = 1.381 e^{-5} \frac{T}{273.15}^{1.81} \left(1 - \frac{a}{1000}\right) \text{,} \tag{9}$$

where $T$ (K) is soil temperature. Derived estimates of $\tilde{v}_{inv}$ were then used to calculate $k_{iso}$, by replacing $v_{inv}$ with $\tilde{v}_{inv}$ in Eq. 2. The difference between the semi-infinite and finite-depth solutions are shown in Fig. S2 for different soil depths. For this soil type and water content, a soil depth of > 8 cm is necessary to be able to neglect the influence of boundary conditions found at the bottom of the incubation vessel.

Corrections for a finite soil depth had also to be applied to estimate $\delta_{eq}$. For this, the soil-depth adjusted isotopic fractionation associated with the diffusion of $CO_2$, $\tilde{a}$, was first calculated as:

$$\tilde{a} = a \left(1 - \frac{\kappa \phi_a D}{\tilde{v}_{inv} z_{max}} \tanh\left(\frac{\tilde{v}_{inv} z_{max}}{\kappa \phi_a D}\right)\right) \text{,} \tag{10}$$

where $a$ was set to 8.8 ‰ (Riley, 2005). Estimates of $\delta_{eq}$ were then obtained using the intercept $c$ of the linear regression and by replacing $m$ with $-\tilde{v}_{inv} C_a / F_R$ and $a$ with $\tilde{a}$ in Eq. 3b. To allow for comparison with $\delta_{sw,ce}$, derived estimates of $\delta_{eq}$ were converted to equivalent values of $\delta_{sw,eq}$ based on the temperature-dependant equilibration fractionation between water and $CO_2$ and the difference between the $VPDB_g$ and VSMOW-SLAP scales (Brenninkmeijer et al., 1983; Kapiluto et al., 2007; Wingate et al., 2010):

$$\delta_{sw,eq} = \delta_{eq} + 0.2 (T - 297.15) \text{,} \tag{11}$$

Treatment summaries are reported as mean and standard deviation unless stated otherwise. Differences in $\delta_{sw,eq}$ and $k_{iso}$ among $\delta_{iw}$ treatments, with significance reported at $p < 0.01$, were tested through a one-way analysis of variance with post-hoc comparison by Tukey's HSD (Mendiburu, 2016).

## 3 Results

### 3.1 Soil properties

The GWC and pH of freshly sampled, sieved soil was $0.207 \pm 0.0054$ g g$^{-1}$ and $4.75 \pm 0.053$ respectively. The pH of the air-dried soil was $4.34 \pm 0.016$. Soil properties were similar among $\delta_{iw}$ treatments. Following drying and storage, the GWC of air dried soil measured during incubation preparation was $0.00993 \pm 0.0012$ g g$^{-1}$. After re-wetting and mixing, the GWC of the soil placed into incubation chambers was $0.107 \pm 0.0018$ g g$^{-1}$. Given 300 g of wet soil placed into each incubation, chambers contained $271 \pm 0.45$ g of dry soil. During the 24-hour pre-incubation $0.172 \pm 0.075$ g of water evaporated and a



further 0.361 ± 0.05 g of water was subsequently lost over the course of the 96 min gas exchange measurement. The $z_{max}$ measured following incubation was 60.7 ± 1.3 mm. The GWC between 0 and 5 cm were 0.0992 ± 0.0048 g g$^{-1}$ at 0–1 cm, 0.105 ± 0.0025 g g$^{-1}$ at 1–2 cm, 0.106 ± 0.0031 g g$^{-1}$ at 2–3 cm, 0.107 ± 0.003 g g$^{-1}$ at 3–4 cm, and 0.107 ± 0.0024 g g$^{-1}$ at 4–5 cm. Bulk density was 1.06 ± 0.023 g cm$^{-3}$ and taking into account the effect of water losses, $\theta_w$ at the mid-point between the start and end of incubations was 0.111 ± 0.0024 m m$^{-3}$. Subsequently, $\phi_t$, $\phi_a$, and $\kappa$ were 0.602 ± 0.0088, 0.491 ± 0.011 and 0.571 ± 0.01, respectively. Values of $z_{max}$, $\phi_t$ and $\theta_w$ carried forward to calculate model solutions are summarised by $\delta_{iw}$ treatment in Table 1.

### 3.2 Water composition

The $\delta_{sw,ce}$ of freshly sampled soil was -3.63 ± 0.10 ‰ VSMOW-SLAP. After air drying and storage, the $\delta_{sw,ce}$ of the residual water pool was -6.69 ± 0.01 ‰ VSMOW-SLAP. Mean $\delta_{iw}$ and standard errors for $\delta_{iw,low}$, $\delta_{iw,med}$ and $\delta_{iw,high}$ irrigation waters were respectively -6.74 ± 0.03, -3.69 ± 0.03 and 0.24 ± 0.04 ‰ VSMOW-SLAP. The addition of these waters to the air-dried soils used in the incubations resulted in initial $\delta_{sw,ce}$ of -7.03 ± 0.34, -4.28 ± 0.21 and -0.795 ± 0.049 ‰ VSMOW-SLAP for the $\delta_{iw,low}$, $\delta_{iw,med}$ and $\delta_{iw,high}$ treatments, respectively. The difference between $\delta_{iw}$ and the initial $\delta_{sw,ce}$ was 0.29 ± 0.35 ‰ VSMOW-SLAP for the $\delta_{iw,low}$ treatment, 0.58 ± 0.22 ‰ VSMOW-SLAP for the $\delta_{iw,med}$ treatment and 1.04 ± 0.05 ‰ VSMOW-SLAP for the $\delta_{iw,high}$ treatment. Evaporation during pre-incubation and gas measurements (Fig. 3) resulted in final $\delta_{sw,ce}$ (averaged between 0 and 5 cm depth and weighted by water content) of -6.75 ± 0.11, -4.17 ± 0.36 and -0.55 ± 0.065 ‰ VSMOW-SLAP for the $\delta_{iw,low}$, $\delta_{iw,med}$ and $\delta_{iw,high}$ treatments, respectively.

The $\delta_{sw,ce}$ of the methodological tests using soil where residual water was removed prior to labelling by oven drying was -5.41 ± 0.19 ‰ VSMOW-SLAP, i.e., 0.57 ‰ more depleted than the labelling water. Similarly, the $\delta_{sw-ce}$ of the soil where residual water was removed by cryogenic extraction ($n = 5$) was -5.01 ± 0.18 ‰ VSMOW-SLAP which overlapped the labelling water with a mean depletion of only 0.17 ‰.

### 3.3 Gas measurements

The mean $CO_2$ concentration $C_b$ of the gas mixtures, $\delta_{b,low}$, $\delta_{b,med}$, and $\delta_{b,high}$, measured in the bypass line were between 422 and 426 ppm for all water treatments, with standard deviations for individual gas mixtures and water treatments smaller than 2 ppm (Table 2). The mean $CO_2$ concentration $C_a$ measured in the chamber line for these inlet conditions varied between 480 and 497 ppm, with standard deviations for individual gas mixtures and water treatments between 2.9 and 15 ppm (Table 2). Subsequently, the resultant $CO_2$ fluxes $F_R$ (Eq. 5) for $\delta_{b,low}$, $\delta_{b,med}$, and $\delta_{b,high}$ were 1.98 ± 0.01 µmol m$^{-2}$ s$^{-1}$ for the $\delta_{iw,low}$ treatments, 2.51 ± 0.01 µmol m$^{-2}$ s$^{-1}$ for the $\delta_{iw,med}$ treatments and 2.22 ± 0.01 µmol m$^{-2}$ s$^{-1}$ for the $\delta_{iw,high}$ treatments, with standard deviations for individual gas mixtures and water treatments between 0.1 and 0.52 µmol m$^{-2}$ s$^{-1}$. The similarity





within water treatments demonstrates the good repeatability of the experiment, however the greater variability in the concentration of $CO_2$ in the chamber $C_a$ compared to that in the bypass line $C_b$ indicates that steady-state conditions may not have been fully attained in all experiments (the order in which the gas mixtures were introduced to the inlet of the chamber was varied between measurement sequences). Indeed whilst there was no general relationship between $F_R$ and $C_b$, $F_R$ linearly

decreased with time ($r^2$ = 0.83 to 0.99), at a rate between 0.078 to 0.31 µmol m$^{-2}$ h$^{-1}$ over the course of all the incubations. Because of these trends, the mean ratios $C_a/F_R$ were 10200 ± 1200, 8350 ± 1100 and 9140 ± 320 s m$^{-1}$ for the $\delta_{iw,low}$, $\delta_{iw,med}$, and $\delta_{iw,high}$ treatments, respectively. Standard deviations associated with the ratio for an individual incubation were between 240 and 425 s m$^{-1}$.

Mean $\delta_b$ and $\delta_a$ values for the different treatments are given in Table 2 together with the resultant $\delta_R$ values (Eq. 6). Unlike $F_R$ there was no general relationship between $\delta_R$ and time but, as expected, $\delta_R$ was strongly negatively correlated with the $\delta_b$ value of the bypass line ($r^2$ > 0.95 in all the incubations). Generally variability in these measurements was largest for the $\delta_{b,low}$ gas mixture.

### 3.4 Estimates of $k_{iso}$ and $\delta_{sw,eq}$

In all the incubations $\delta_a$ and $\delta_R$ were strongly ($r^2$ > 0.93) and negatively correlated (Fig. 4). The slope $m$ of the linear regressions were broadly similar among water treatments with means of -1.83 ± 0.25 for the $\delta_{iw,low}$ treatment, -1.32 ± 0.25 for the $\delta_{iw,med}$ treatment and -1.53 ± 0.15 for the $\delta_{iw,high}$ treatment. In contrast, the intercept $c$ of the linear regressions between $\delta_a$ and $\delta_R$ were distinct with treatment means of -30.2 ± 2.1 ‰ VPDB$_g$ for the $\delta_{iw,low}$ treatment, -20.1 ± 2.7 ‰ VPDB$_g$ for the $\delta_{iw,med}$ treatment and -14.3 ± 1.2 ‰ VPDB$_g$ for the $\delta_{iw,high}$ treatment.

The mean piston velocity assuming a semi-infinite soil column $v_{inv}$ (Eq. 3a) varied between 0.16 and 0.18 mm s$^{-1}$ (Table 3). Accounting for the finite depth of the soil column lead to $\tilde{v}_{inv}$ values (Eq. 7) that were systematically but only marginally larger (Table 3), mostly because soil depth had been chosen to minimise this difference (Table 1 and Fig. S2). These values led to $CO_2$-$H_2O$ isotopic exchange rates $k_{iso}$ that were not significantly different between treatments, with means of 0.08 ±
0.01, 0.06 ± 0.02 and 0.07 ± 0.01 s$^{-1}$ for $\delta_{iw,low}$, $\delta_{iw,med}$, and $\delta_{iw,high}$, respectively (Table 3).

Estimates of $\tilde{a}$ (Eq. 10) were approximately half the full fractionation of 8.8 ‰ (Table 3). Expectedly, there were significant differences in $\delta_{eq}$ (Eq. 3b) estimates among water irrigation treatments, with values of -8.7 ± 0.2 ‰ VPDB$_g$ for the $\delta_{iw,low}$ treatment, -6.44 ± 0.52 ‰ VPDB$_g$ for the $\delta_{iw,med}$ treatment and -3.56 ± 0.18 ‰ VPDB$_g$ for the $\delta_{iw,high}$ treatment. These values
led to equivalent values of $\delta_{sw,eq}$ (Eq. 11) that were also significantly different between water irrigation treatments, and surprisingly more depleted than the isotopic composition of cryogenically extracted soil water $\delta_{sw,ce}$ at all depths (Fig. 3).



## 4 Discussion

Our first aim was to confirm the assumption that the isotopic compositions of the $CO_2$ flux ($\delta_R$) and the $CO_2$ of the air at the soil-air interface ($\delta_a$) are linearly related (Eq. 1). This appears to be a good approximation for our data with strong correlations between $\delta_R$ and $\delta_a$ in all experiments (Fig. 4). However, a number of other assumptions inherent to applying the

model described by Eq. 1 in this way may influence our results. Namely, that $CO_2$ production profiles were constant with depth and that gas flux measurements were made under steady-state conditions (Tans, 1998). The first point is unlikely to be an issue here because care was taken to homogenise the soils before each gas exchange measurement (see Methods) and the total period between preparation and measurement (24 hr) was too short to allow large gradients, for example in soil moisture, that might cause an unequal respiration profile to develop. Potential deviations from steady-state conditions require

more attention. A period of 21 min was included before the measurements from which fluxes were calculated, not only at the initial connection of an incubation chamber but also after each subsequent switch of the inlet conditions (Fig. 2). This period was chosen based on initial tests indicating that 10 min was sufficient for the concentration and composition of $CO_2$ to stabilise in an empty chamber and calculations indicating that full isotopic equilibrium of the $CO_2$-$H_2O$ system, given the acidity and temperature of the soil, should theoretically be reached within *ca.* 12 minutes, even in the absence of any

catalysed reaction (Uchikawa and Zeebe, 2012). However, our results showed that the soil $CO_2$ flux $F_R$ systematically decreased with time, by about 10 % on average between the first and last measurements within an incubation. This indicates that our measurements did not strictly adhere to the assumption of steady state. This trend probably reflects a combination of dissolved $CO_2$ de-gassing from the soil water as a result of small differences between pre-incubation and incubation $CO_2$ concentrations and, probably more importantly, the temporal response of soil respiration to re-wetting (Birch, 1958; Jarvis et

al., 2007). As such, the ratio between $C_a$ and $F_R$ is not strictly constant and this introduces uncertainties into the estimations of $k_{iso}$ and $\delta_{sw-eq}$, that are considerably larger than that associated with individual gas measurements. Indeed, propagating this within-incubation variability in the ratio $C_a/F_R$ leads to within-incubation uncertainties in $k_{iso}$ of around 0.01 s$^{-1}$ for all treatments and within-incubation uncertainties in $\delta_{sw-eq}$ of 0.45, 0.37 and 0.24 ‰ for the $\delta_{iw,low}$, $\delta_{iw,med}$ and $\delta_{iw,high}$ treatments. These within-incubation uncertainties are the same of order magnitude as the variability in treatment means (Table 3),

indicating that this temporal deviation from steady-state conditions may contribute to the majority of the reported variability. This variability could be reduced in future studies by working with fresh soils or allowing longer acclimatisation periods after re-wetting, provided efforts are made to limit the development of heterogeneity in the soil profile. As the linearity between $\delta_R$ and $\delta_a$ appears strong, variability could also be reduced by considering two rather than three combinations of $\delta_b$ to shorten the overall measurement time.



Secondly, we aimed to compare estimates of $\delta_{sw,eq}$ determined from the gas flux measurements with $\delta_{sw,ce}$ measured for the extracted bulk soil water. Whilst $\delta_{sw,eq}$ estimates were distinct among water treatments, they were also consistently more depleted than $\delta_{sw,ce}$ with mean offsets from $\delta_{sw,ce}$ between 0 and 5 cm of -2.56 ± 0.11, -2.87 ± 0.56 and -3.61 ± 0.23 ‰ VSMOW-SLAP for the $\delta_{iw,low}$, $\delta_{iw,med}$, and $\delta_{iw,high}$ treatments, respectively. Given the difference between the externally assigned value of -3.45 ± 0.02 ‰ $VPDB_g$ for $\delta_{b,high}$ and its measurement, with treatment means of -3.63 ± 0.15, -3.62 ± 0.3 and -3.58 ± 0.11 ‰ $VPDB_g$ for $\delta_{iw,low}$, $\delta_{iw,med}$, and $\delta_{iw,high}$ respectively, we find no evidence that an offset between the calibration of our gas and water measurement scales could account for the size of the offset observed. Similarly, we find no evidence for large biases in our cryogenic extraction methodology as $\delta_{sw,ce}$ and composition of the labelling water overlapped in the tests where residual water was removed under extraction conditions prior to label addition (Orlowski et al., 2013, 2016b) . This suggests that the offset between $\delta_{sw,eq}$ and $\delta_{sw,ce}$ is the result of $CO_2$ interacting with a water pool whose isotopic composition is more depleted than $\delta_{sw,ce}$. Differences in the water pools characterised by different methodologies for determining the isotopic composition of soil waters are well known, with the cryogenic extraction method being expected to remove mobile, immobile, hygroscopic and potentially crystalline water, whilst the static equilibration of soils with $CO_2$ is expected to principally reflect only the mobile and immobile pools (Hsieh et al., 1998b; Orlowski et al., 2016b; Sprenger et al., 2015). For this reason, we might expect $\delta_{sw,eq}$ derived here to reflect the isotopic composition of mobile and immobile pools and $\delta_{sw,ce}$ to reflect both these as well as hygroscopic and crystalline waters. However, disregarding the crystalline water pool in this predominately sandy soil, this would suggest that the isotopic composition of the hygroscopic pool would have to be considerably more positive than that of the mobile and immobile water pools in order to account for the fact that $\delta_{sw,eq}$ estimates were more negative than $\delta_{sw,ce}$. Contrary to this, extraction of water from the air-dried soil, of which hygroscopic water presumably represents a greater portion of the total water pool than in re-wetted samples, yielded a $\delta_{sw,ce}$ more depleted than that of the freshly sampled soil and the initial compositions of both the $\delta_{iw,med}$, and $\delta_{iw,high}$ treatments. This suggests that this residual pool in fact reflects the potentially rapid and dynamic exchange with atmospheric vapour during storage (Lin and Horita, 2016; Orlowski et al., 2016b; Savin and Epstein, 1970). Indeed, the fact that about 10 % of water in the re-wetted soils consisted of this residual pool helps account for the differences between $\delta_{iw}$ and initial $\delta_{sw-ce}$ across the water treatments. Observations that fractionation occurs between water associated with cations and anions present in solution or mineral surfaces (Oerter et al., 2014), organic particle surfaces (Chen et al., 2016) and pore spaces (Lin and Horita, 2016) and bulk water pools may help explain our data. In particular, Chen et al. (2016) found that the composition of the water bound to organic particle surfaces could be up to 4 ‰ more depleted than bulk water, with a greater fraction at higher volumetric ratios of solid to water. Such a depletion in bound (hygroscopic) water would explain our $\delta_{sw,ce}$ data, given the relatively high carbon content and low water content of the soil studied. However this requires us to consider that $CO_2$ is not principally equilibrating with mobile and immobile water but rather with hygroscopic water. Interaction between $CO_2$ and such surface pools may be plausible if this is where microbial communities expressing CA are present and active. If this were the case we



should expect to see a smaller offset between $\delta_{sw,eq}$ and $\delta_{sw,ce}$ at higher water contents as the relative influence of surface pools decrease (Chen et al., 2016; Hsieh et al., 1998b; Oerter et al., 2014). We tested this by creating and measuring an additional six incubations, following the methods described above, that were re-wetted with different amounts of the same water to achieve water contents ranging from about 15 to 70 % water-filled pore space. The difference between estimated $\delta_{sw,eq}$ and

$\delta_{sw,ce}$ determined for sampling depths of 0-1 cm ($r^2 = 0.92$) and 1-5 cm ($r^2 = 0.88$) were positively and linearly correlated with water-filled pored space across these incubations (Fig. 5). The fact that these relationships indicate the offset is smallest around 95 % water-filled pore space may indeed support the inference that estimates of $\delta_{sw,eq}$ are being influenced by fractionations between surface and bulk water pools.

Finally we aimed to compare the sensitivity of estimates of $k_{iso}$ to $\delta_{sw}$. Estimates of $k_{iso}$ ranged from 6.8 to 14.6 times greater than the theoretical un-catalysed rate $k_{iso,uncat}$ of 0.0062 s$^{-1}$ for the incubation conditions (Uchikawa and Zeebe, 2012). This enhancement of the apparent over the un-catalysed rate of hydration indicates that the reaction is indeed being catalysed by the presence of CA in these soils. After considering differences in the value of $k_{iso,uncat}$ used, these rates are on the lower end of those reported from soil chambers deployed in the forest where soil used here was collected (Wingate et al., 2009; Wingate et

al., 2010).  Our estimates of $k_{iso}$ were not distinct among water treatments (Table 3), supporting the idea that our new approach is robust enough to assay soil CA activity in the absence of information about $\delta_{sw}$. This is beneficial as this approach also does not rely on any assumption about which soil water pool the $CO_2$ must equilibrate with, as such an assumption could introduce strong biases in the retrieved $k_{iso}$. To illustrate this point, we compared our new approach with previous studies (Wingate et al., 2009, 2008) where $\delta_{sw}$ has been used to determine $k_{iso}$ from Eq. 1 and Eq. 2. For this, we

replaced $v_{inv}$ and $a$ in these equations with $\tilde{v}_{inv}$ (Eq. 7) and $\tilde{a}$ (Eq. 10) and iteratively solved for $\tilde{v}_{inv}$ using $\delta_{sw,ce}$ and gas measurements made at the individual $\delta_b$ conditions reported here. In doing so we highlight the potential sensitivity of estimating $k_{iso}$ in this way to the combination of *both* $\delta_{sw}$ and $\delta_b$ conditions. Equivalent values of $\delta_{eq}$ calculated (Eq. 11) from $\delta_{sw,ce}$ between 0 and 5 cm were -6.15 ± 0.11, -3.57 ± 0.36 and 0.049 ± 0.065 ‰ VPDB$_g$ for the $\delta_{iw,low}$, $\delta_{iw,med}$, and $\delta_{iw,high}$ treatments, respectively. Treatment means for $k_{iso}$ estimated in this way using measurements made at $\delta_{b,low}$ were 0.0424 ±

0.0045, 0.029 ± 0.0078 and 0.035 ± 0.0034 s$^{-1}$ for the $\delta_{iw,low}$, $\delta_{iw,med}$, and $\delta_{iw,high}$ treatments, respectively. These rates are roughly half the size of $k_{iso}$ estimated from regression among multiple $\delta_b$ conditions (Table 3). Positive solutions for $k_{iso}$ estimated using measurements made at $\delta_{b,med}$ were not found for all incubations and estimates were smaller still with treatment means of 0.0059 ± 0.0021, 0.0079 ± 0.0051 and 0.015 ± 0.0042 s$^{-1}$ for the $\delta_{iw,low}$ ( $n = 3$), $\delta_{iw,med}$ ( $n = 5$) and $\delta_{iw,high}$ ( $n = 6$) treatments, respectively. No solution was found for $k_{iso}$ estimated using measurements made at $\delta_{b,high}$ for the $\delta_{iw,med}$ and $\delta_{iw,high}$ water

treatments, whilst the estimates for $\delta_{iw,low}$ water treatment were two orders of magnitude larger with a mean of 3.18 ± 2.6 s$^{-1}$, i.e., more than 500 times larger than $k_{iso,uncat}$. Situations were no solution are found arise in cases where $(\delta_{eq} - \delta_R)(\delta_{eq} - \delta_a) \geq 0$. These cases and the considerable variability in $k_{iso}$ among water treatments under different $\delta_b$ conditions reflect the presence



of an asymptote in the model response when $\delta_{eq}$ and $\delta_a$ are similar (Eq. 1). In this region of the response, relatively small changes in $\delta_{eq}$ result in large changes in $k_{iso}$ as seen among estimates for the $\delta_{iw,low}$ water treatment made using $\delta_{b,low}$, $\delta_{b,med}$ and $\delta_{b,high}$ conditions (Fig. 6). From this analysis we can see that this can be problematic when attempts to estimate $k_{iso}$ are made under field conditions. The oxygen isotopic composition of atmospheric $CO_2$ ($\delta_{atm}$) typically ranges from -1.5 to +1.5 ‰ VPDB$_g$ (Welp et al., 2011). The isotopic composition of superficial soil water varies as a function of latitudinal and altitudinal patterns in the composition of precipitation and the subsequent influence of evapotranspiration, but is frequently reported in the range of -10 to +5 ‰VSMOW (Barbeta et al., 2015; Brooks et al., 2010; Dawson et al., 2002; Hsieh et al., 1998a). This suggests that values of $\delta_{eq}$, after accounting for the influence of soil temperature and difference between scales in converting from $\delta_{sw}$ (Eq.11), can overlap $\delta_{atm}$ under normal conditions (Wingate et al., 2009).

## 5 Conclusions

We have demonstrated that a strong linear correlation existed between $\delta_R$ and $\delta_a$, so that we should also be able to derive soil CA activity ($k_{iso}$) and isotopic composition ($\delta_{eq}$) from only two inlet conditions. Given the temporal changes seen in the net flux after even 24h soil irrigation, this is likely to decrease even further the uncertainty in the estimation of $k_{iso}$.

We also reported an offset between $\delta_{sw,ce}$ and $\delta_{sw,eq}$ which is an indication that bulk soil water may not be the pool of interaction with $CO_2$. Such an offset might be explained by an isotopic fractionation between bound and mobile water pools while the sign of this offset seems to indicate that $CO_2$ interacts preferentially with bound water. Clearly, a better understanding of the fine-scale heterogeneity in soil water isotope composition and microbial activity is required.

Finally, our estimates of $k_{iso}$ were independent of the isotopic composition of the irrigation water used, suggesting our approach is a robust assay of the activity, even in soils with low CA activities as reported here. Given the sensitivity of $k_{iso}$ when estimated using a single $CO_2$ composition and prescribed vales of $\delta_{sw,eq}$, our approach clearly represents a more conservative and robust method. To better understand the cycling of oxygen isotopes of $CO_2$ within soils, further work is required to understand the physical processes controlling fractionation between soil water pools and the relation of microbial communities and their enzymes to these pools.



## Author contributions

S.J, J.O, J.S, T. L, A. B. & L.W. developed the modelling and data processing approach.

S.W., J.S. and S.J. designed and built the gas exchange system.

S.J. and S.W.  conducted gas flux measurements.

5   N.S , N.F, J.M, and S.J. tested the water extraction system and analysis system and conducted the soil water extractions and water analysis.

S.J. wrote the manuscript with contributions from all authors.

## Acknowledgements

We would like to thank Régis Burlett, Callum Tyler, Thomas Sajus, and Jason West for their input in the laboratory and
10  development of the experiment. This project has received funding from the European Research Council (ERC) under the European Union's Seventh Framework Programme (FP7/2007-2013) (grant agreement No. 338264) awarded to L.W. Salary for J.S. was funded by the Agence Nationale de la Recherche (ANR, project ORCA) and the INRA departments EA and EFPA.





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





## Tables

Table 1: Soil properties for the different water irrigation treatments, *i.e.*, maximum soil depth ($z_{max}$), total porosity ($\phi_t$) and volumetric soil moisture content ($\theta_w$). Means ($n = 6$) and their standard deviations (in parentheses). Lower-case letters indicate significant differences (one-way analysis of variance and Tukey's HSD, $p < 0.01$) among water treatments.

| Treatment | $z_{max}$ (mm) | $\phi_t$ | $\theta_w$ |
|---|---|---|---|
| $\delta_{iw,low}$ | 60.3 (1.5) a | 0.599 (0.01) a | 0.112 (0.002) a |
| $\delta_{iw,med}$ | 61.5 (0.76) a | 0.607 (0.0043) a | 0.11 (0.0023) a |
| $\delta_{iw,high}$ | 60.3 (1.4) a | 0.599 (0.0093) a | 0.112 (0.0024) a |





Table 2: Gas exchange data for the different irrigation treatments ($\delta_{iw,low}$, $\delta_{iw,med}$ and $\delta_{iw,high}$) and the different $CO_2$ isotope composition on the bypass line ($\delta_{b,low}$, $\delta_{b,med}$ and $\delta_{b,high}$). Means ($n = 6$) and their standard deviations (in parenthesis).

| | $\delta_{iw,low}$ | | | $\delta_{iw,med}$ | | | $\delta_{iw,high}$ | | |
|---|---|---|---|---|---|---|---|---|---|
| | $\delta_{b,low}$ | $\delta_{b,med}$ | $\delta_{b,high}$ | $\delta_{b,low}$ | $\delta_{b,med}$ | $\delta_{b,high}$ | $\delta_{b,low}$ | $\delta_{b,med}$ | $\delta_{b,high}$ |
| $C_b$ | 426 | 424 | 425 | 424 | 423 | 425 | 426 | 422 | 425 |
| (ppm) | (1.4) | (0.92) | (0.036) | (1.7) | (0.99) | (0.14) | (1.5) | (0.83) | (0.031) |
| $C_a$ | 482 | 480 | 481 | 495 | 494 | 497 | 489 | 485 | 488 |
| (ppm) | (9.8) | (8.6) | (6.4) | (11) | (13) | (15) | (3.9) | (3.5) | (2.9) |
| $F_R$ | 1.98 | 1.99 | 1.98 | 2.51 | 2.51 | 2.53 | 2.22 | 2.21 | 2.22 |
| ($\mu$mol m$^{-2}$ s$^{-1}$) | (0.31) | (0.3) | (0.23) | (0.43) | (0.42) | (0.52) | (0.17) | (0.12) | (0.099) |
| $\delta_b$ | −26.8 | −13.9 | −3.63 | −26.9 | −14 | −3.62 | −26.9 | −14 | −3.58 |
| (‰ VPDB$_g$) | (0.25) | (0.092) | (0.15) | (0.22) | (0.17) | (0.3) | (0.062) | (0.082) | (0.11) |
| $\delta_a$ | −22.5 | −13 | −5.56 | −21.9 | −12.4 | −5.11 | −21.1 | −11.6 | −4.2 |
| (‰ VPDB$_g$) | (0.26) | (0.1) | (0.19) | (0.34) | (0.094) | (0.51) | (0.23) | (0.1) | (0.17) |
| $\delta_R$ | 10.8 | −6.04 | −20.3 | 8.33 | −2.67 | −14.0 | 17.7 | 4.4 | −8.36 |
| (‰ VPDB$_g$) | (3.7) | (1.2) | (0.95) | (3.8) | (1.1) | (1.3) | (2.0) | (1.2) | (0.74) |





Table 3: Model solutions for the different water irrigation treatments. Means ($n = 6$) and their standard deviations (in parentheses). Lower-case letters indicate significant differences (one-way analysis of variance and Tukey's HSD, $p < 0.01$) among water treatments.

| Treatment | $v_{inv}$ (mm s$^{-1}$) | $\tilde{v}_{inv}$ (mm s$^{-1}$) | $k_{iso\text{-}ap}$ (s$^{-1}$) | $\tilde{A}$ (‰ VPDB$_g$) | $\delta_{sw,eq}$ (‰ VSMOW) |
|---|---|---|---|---|---|
| $\delta_{iw,low}$ | 0.179 (0.011) a | 0.181 (0.011) a | 0.0799 (0.0086) a | 5.36 (0.16) a | -9.31 (0.2) c |
| $\delta_{iw,med}$ | 0.158 (0.021) a | 0.162 (0.02) a | 0.0633 (0.015) a | 4.88 (0.43) a | -7.04 (0.52) b |
| $\delta_{iw,high}$ | 0.168 (0.014) a | 0.171 (0.013) a | 0.0712 (0.012) a | 5.14 (0.29) a | -4.16 (0.18) a |

# Figures

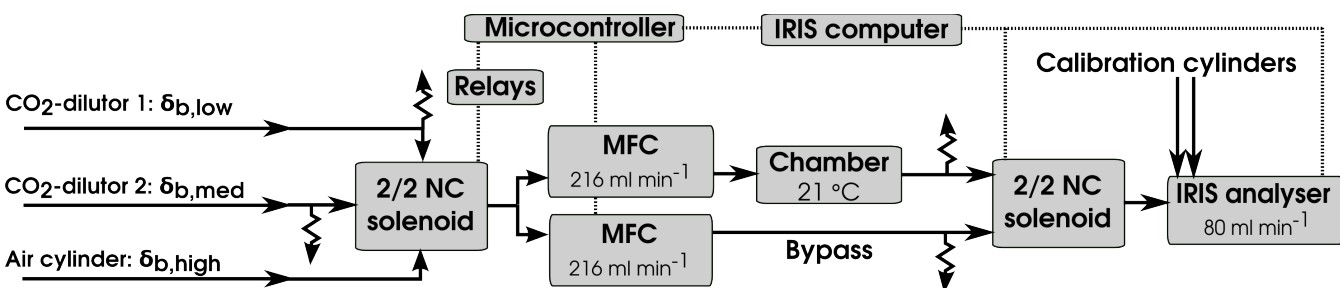

Figure 1: Schematic of the system used to make gas exchange measurements.





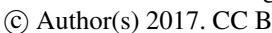

Figure 2: An example of the gas exchange measurement sequence, scanning sequentially calibration cylinders, the chamber line during a stabilisation period, calibration cylinders again, and finally the chamber ($C_a$, $\delta_a$) and bypass ($C_b$, $\delta_b$) lines, for three $CO_2$ isotopic composition values of the bypass line ($\delta_b$). In this case, the $\delta_b$ inlet conditions, whose changes are indicated by the vertical dashed lines, started with $\delta_{b,med}$ and ended with $\delta_{b,low}$. (a) Total $CO_2$ concentration and (b) the $\delta^{18}O$ of $CO_2$.




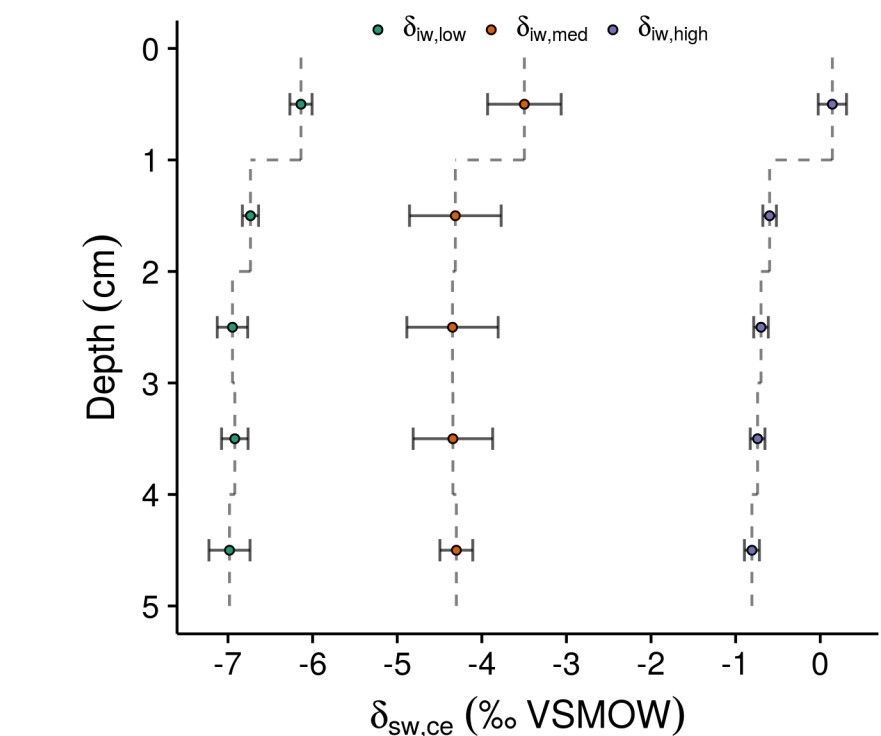

Figure 3: Profiles of $\delta_{sw,ce}$ cryogenically extracted from incubated soils at intervals of 0-1, 1-2, 2-3, and 4-5 cm.



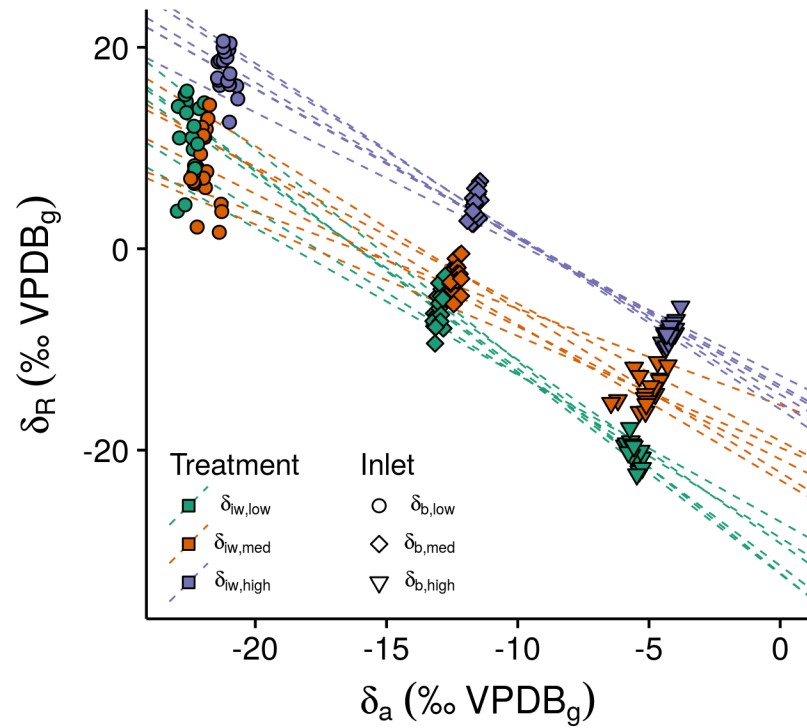

Figure 4: Relationships between $\delta_R$ and $\delta_a$ for the different irrigation water treatments. Dashed lines indicate linear regressions for individual incubations.



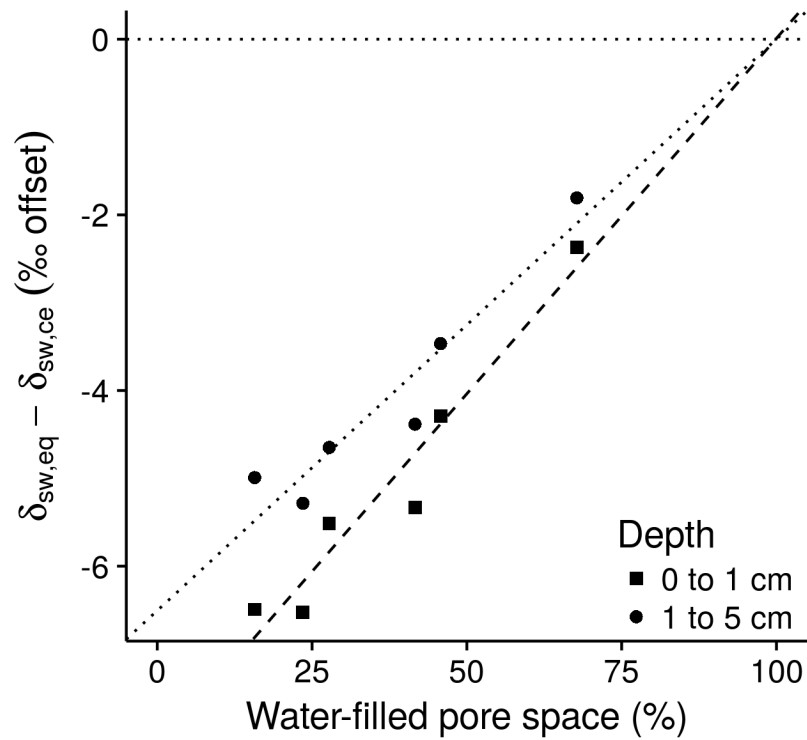

Figure 5: Relationships between water-filled pore space and the difference between estimates of $\delta_{sw,eq}$ and $\delta_{sw,ce}$ determined for depths of 0-1 cm and 1-5 cm. Dashed lines indicate linear regressions for the two sampling depths.



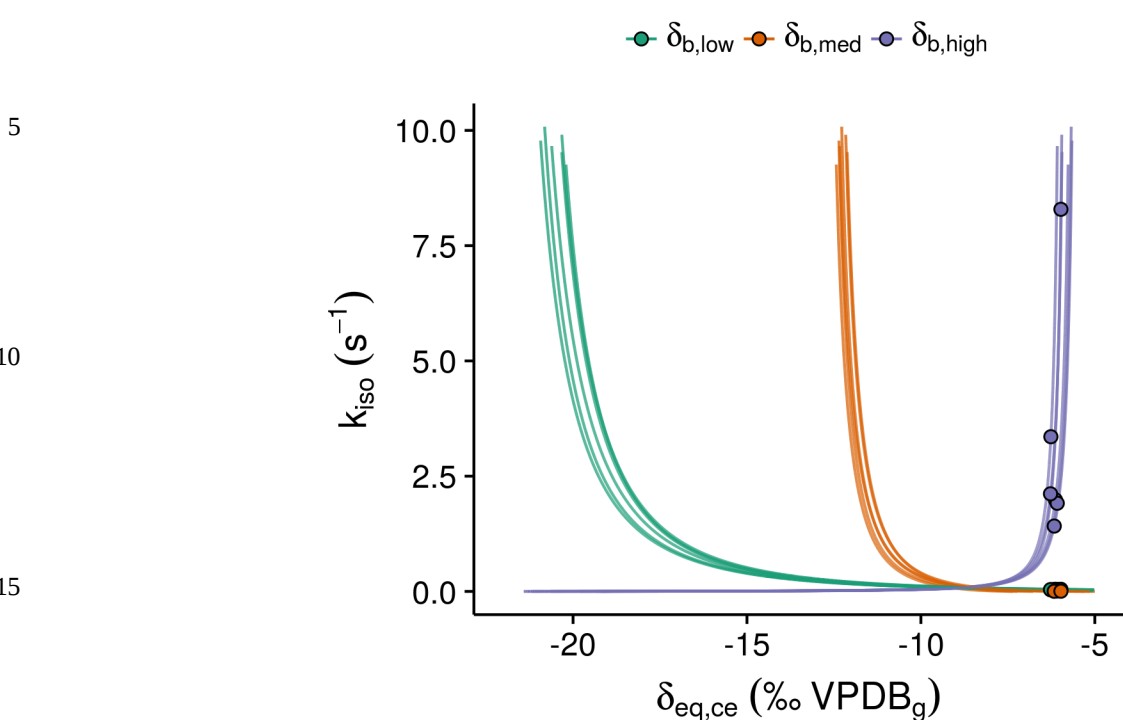

Figure 6: Model responses between $k_{iso}$ and $\delta_{eq}$ at different $\delta_b$ conditions for $\delta_{iw,low}$ water treatments.