# Peer review of "Non-destructive estimates of soil carbonic anhydrase activity and associated soil water oxygen isotope composition"

_Hydrology and Earth System Sciences, 2017_

## Referee Comment (RC1) · M. Sprenger (Referee) · 30 Aug 2017

The manuscript "Non-destructive estimates of soil carbonic anhydrase activity and soil water oxygen isotope composition" by Jones et al. describes in detail their suggested method to estimate the activity of carbonic anhydrase in soils using the $\delta^{18}O$ signal in CO2 soil vapor flux. The manuscript is very well prepared with a detailed description in the methods section and a good presentation of their findings in tables and figures. The study is of general interest to the readership of Hydrology and Earth System Science and I recommend publication after addressing the comments below.

General comments

To my understanding, the non-destructive soil water $\delta^{18}O$ estimations would be either limited to an integrated signal (no depth information) or would need to be conducted with in-situ devices sampling the soil vapor. Thus, the title is a bit misleading as one is often interested in the depth information of the soil water isotope composition. I think that this lack of depth information should be also discussed in the manuscript.

I do not agree with the interpretation of Figure 5 that $\delta_{sm,eq}$ is in equilibrium with waters in hygroscopic water (see P14 L31). Given that the difference between $\delta_{sm,eq}$ and $\delta_{sm,ce}$ is smallest for wettest soils reveals the opposite: The wetter the soil, the smaller is the ratio between volumes of soil water in soil pores and volume of waters in soil pores plus hygroscopic waters. If equilibration would preferably take place with the hygroscopic water, the differences should be highest for wetter soil, as the hygroscopic water would become small relative to the bulk pore water volume (Figure 1).

[Figure]

*Figure 1 Upper panel: Volume of water that is hygroscopic bound (red) and that is not hygroscopic bound (blue) as a function of the water filled pore space. Lower panel: The ratio between both as a function of the water filled pore space.*

Specific comments:

P1 L30: As outlined above, I do not understand that conclusion as I would interpret the Figure 5 differently.

P4 L8: I am missing a clear research question here. You present the research gap, but do not state any hypothesis or research question before getting to the objectives in L9ff.

P12 L16: I suggest providing statistical tests rather than using "broadly". Also for the L19 "distinct".

P12 L31: You do not present $\delta_{sm,eq}$ in the Figure 3. Please add.

P14 L14: I do not like "immobile" water pool and encourage to use a different term, as the soil water held at low pressure heads is less mobile, but not stagnant. However, I know that this is widely used and common nomenclature is missing. Maybe "less mobile" or "water at lower pressure heads"? Or instead "mobile and immobile" using "bulk soil water"?

P15 L2-L6: This reads more like results and introducing a new figure would also better fit to the results section.

P15 L7: You do not have a data point at 95% water filled pore space. Therefore, I prefer you refer here to 75%.

Table 2: Be consistent with the decimal places for the delta-values.

Table 3: In the 5$^{th}$ column, it should be "ã" not "Ã"

Figure 2: Is the dotted grey line showing the measurements at 1Hz and the dots, diamonds and triangles are showing the average values integrated over time?

Figure 3: Consider adding the $\delta_{sm,eq}$ as you refer to that in the manuscript.

---

## Referee Comment (RC2) · Anonymous Referee #2 · 5 Sep 2017

GENERAL COMMENTS The manuscript "Non-destructive estimates of carbonic anhydrase activity and soil water oxygen isotope composition" by Sam Jones and coauthors reports and discusses a new method to estimate the activity of carbonic anhydrases in soils using isotopic measurements of soil carbon dioxide. The suggested method was tested, for the purpose of this study, using controlled laboratory soil incubations. The objectives of the study are well motivated and explained. The abstract is well-written, clear and informative. The introduction is also well structured, well written and clear, both in terms of the limitations of the currently usually used modeling approach, and about the potential benefits of the new method. The methods are in general explained well and in detail (but see below). Also the discussion is written logical, clear and convincing. I have a few major comments that I would like the authors to consider and address during revision. To start with, the statistical analysis of the data should be improved, and that part of the Methods description should be elaborated and improved. As statistical method to assess the treatment effects in this study I recommend linear mixed effects models, see e.g. [Gueorguieva and Krystal, 2004; Crawley, 2009]. I noted that the reference that is currently used in the Statistics part is missing on the reference list (Mendiburu, 2016). Moreover, the Results section should be improved. In long parts many values are listed, e.g. means and error estimates for several parameters and treatments are spelled out in the text. I suggest to check which values are already given in the Tables, and to consider moving more of the values currently given in the text into Tables to refer to. Also, the authors are using many acronyms throughout the text. I find they are too many and this makes the text in parts hard to read. I suggest to reconsider which acronyms are central and to keep these, but consider to spell out certain variables (i.e. avoid too many acronyms). Alternatively, you might add a list of acronyms to the manuscript and refer to it repeatedly, to facilitate for the reader to look up the meaning of all acronyms during reading. Please check as well that all acronyms are actually defined upon first use, and consider to even define acronyms that are common in your field but may not be obvious to all readers of the article (e.g. VPDBg and VSMOW-SLAP). The same applies to the Tables and Figures, please include in footnotes or legend the meaning of the used acronyms (if you decide to keep them), with the goal that Figures and Tables can be understood independent of the text. As example I refer to the legend of Fig. 6, which contains four acronyms and is difficult to understand in its current form. In Figure 5, please add confidence intervals to the regression lines. This may not be feasible in terms of clarity for Fig. 4, which contains many regression lines in one graph. In that case please add a note to the legend of Fig. 4 why confidence intervals are not shown. In summary, I find this an interesting and important study that is well thought out, conducted, analysed and discussed. However, I recommend major revision of, mostly, the Results section and parts of the Methods section according to above comments, as well as improvement of
the Statistical Methods used for data analysis and its description.

SPECIFIC COMMENTS -P6/L4: Please add a reference for the assumed particle density [Linn and Doran, 1984].

TECHNICAL CORRECTIONS - P1/L11: Move the comma: "..., a group of enzymes that catalyse the hydration of CO2 in soils and plants,..." - P5/20: "were monitored" (change from "was") - P9/L13: "R Development Core Team"

References Crawley, M. J. (2009), The R book, 942 pp., John Wiley & Sons Ltd, Chichester. Gueorguieva, R., and J. H. Krystal (2004), Move over ANOVA, progress in analyzing repeated-measures data and its reflection in papers published in the archives of general psychiatry., Archives of General Psychiatry, 61, 310-317. Linn, D. M., and J. W. Doran (1984), Effect of water-filled pore space on carbon dioxide and nitrous oxide production in tilled and nontilled soils., Soil Science Society of America Journal, 48, 1267-1272.

HESSD

---

## Author Comment (AC1) · 3 Oct 2017

Response to Matthias Sprenger (Referee #1)

We would like to thank Matthias Sprenger for taking the time to review the manuscript. We have reproduced his comments, in blue, along with our responses in below.

To my understanding, the non-destructive soil water $\delta^{18}O$ estimations would be either limited to an integrated signal (no depth information) or would need to be conducted with in-situ devices sampling the soil vapor. Thus, the title is a bit misleading as one is often interested in the depth information of the soil water isotope composition. I think that this lack of depth information should be also discussed in the manuscript.

We agree with this comment, the value of soil water $\delta^{18}O$ estimated does indeed reflect an integrated signal. In addition the depth to which this signal is integrated is hard to define because it is a function of the effective rate of diffusion which controls the residence time of $CO_2$ in the soil profile and the rate of hydration which acts to impart the soil water signature on the $CO_2$. In the case of this study, conducted on shallow soil microcosms with minimal heterogeneity in the soil water content and isotopic composition, the signal likely reflects the influence of the total soil column. We have altered the title to clarify that we refer to the soil water composition associated with hydration of $CO_2$ and the atmospheric signal rather than propose a sensible approach to non-destructively obtain depth-resolved soil water profile data. We have also expanded these points in the text, as detailed below.

Title *"Non-destructive estimates of soil carbonic anhydrase activity and associated soil water oxygen isotope composition"*

P4 L4 *"The appropriate value for $\delta_{eq}$ is then conceptually related to the shallowest depth at which respired or atmospheric $CO_2$ has sufficient time to fully equilibrate with soil water (Miller et al., 1999; Wingate et al., 2009). For example, Wingate et al. (2009) estimate this depth as the soil depth below which $CO^{18}O$ molecules would take more than 4 times longer to diffuse out of the soil than it would take them to re-equilibrate with soil water. However, whilst use of this setting-point is a convenient approximation in field settings (Wingate et al., 2009, 2010), some degree of exchange still occurs above this depth (Kapiluto et al., 2007)."*

P14 L20 *"Given the relatively constant profile of $\delta_{sw,ce}$ with depth (Fig 3) and the fact that total soil depth ($z_{max}$) was shallower than that required for full convergence between the semi-infinite and finite soil depth model solutions (Table 3, Fig S2), the estimates of $\delta_{sw,eq}$ reported likely reflect the influence of interaction between $CO_2$ and soil water across the total soil depth (Kapiluto et al., 2007).*

I do not agree with the interpretation of Figure 5 that $\delta_{sm,eq}$ is in equilibrium with waters in hygroscopic water (see P14 L31). Given that the difference between $\delta_{sm,eq}$ and $\delta_{sm,ce}$ is smallest for wettest soils reveals the opposite: The wetter the soil, the smaller is the ratio between volumes of soil water in soil pores and volume of waters in soil pores plus hygroscopic waters. If equilibration would preferably take place with the hygroscopic water, the differences should be highest for wetter soil, as the hygroscopic water would become small relative to the bulk pore water volume (Figure 1).

We understand and agree with the point made that essentially highlights some deficiency in our explanation. Whilst $CO_2$ appears to be heavily influenced by hygroscopic water in the main experimental tests (conducted at about 20 % WFPS), it is also clear that the proportion of non-hygroscopic to hygroscopic water that $CO_2$ has to interact with increases with water content (Figure 1 in the reviewer comment; future readers please note the y-axis in the lower panel, as drawn, should be Vh/Vnh rather than Vnh/Vh). This occurs because as water content increases, non-hygroscopic water occupies more pore space that $CO_2$ must diffuse through, thus undergoing further hydration and equilibration with this more mobile pool of water. This results in a better agreement between the signal imparted on the $CO_2$ and that of the bulk soil water. We have now clarified this point in the discussion.

P15 L19-L27: *"However this requires us to consider that $CO_2$ is being heavily influenced by exchange with hygroscopic water under our experimental conditions. Such interaction between $CO_2$ and hygroscopic water may be plausible as this is where microbial communities expressing CA are likely to be present and active. If interaction with hygroscopic water were the cause of this observation, we should expect to see a smaller offset between $\delta_{sw,eq}$ and $\delta_{sw,ce}$ at higher water content because, as water content increases, so does the proportion of non-hygroscopic to hygroscopic water that $CO_2$ interacts with during the slow process of liquid phase diffusion (4 orders of magnitude lower than gas phase diffusion). We estimated that, even at the uncatalysed rate of hydration, $CO_2$ molecules would be fully equilibrated if they had to diffuse through about 0.5 mm of water. Whilst this is not realistic for water films adsorbed onto pore-surfaces, such path-lengths are plausible for filled capillaries as the soil-pore network approaches saturation (Lebeau and Konrad, 2010; Tokunaga, 2011; Tuller and Or, 2001)."*

*Lebeau, M. and Konrad, J.-M.: A new capillary and thin film flow model for predicting the hydraulic conductivity of*

*unsaturated porous media, Water Resources Research, 46(12), W12554, doi:10.1029/2010WR009092, 2010.*

*Tokunaga, T. K.: Physicochemical controls on adsorbed water film thickness in unsaturated geological media, Water Resources Research, 47(8), W08514, doi:10.1029/2011WR010676, 2011.*

*Tuller, M. and Or, D.: Hydraulic conductivity of variably saturated porous media: Film and corner flow in angular pore space, Water Resources Research, 37(5), 1257–1276, doi:10.1029/2000WR900328, 2001.*

P1 L30: As outlined above, I do not understand that conclusion as I would interpret the Figure 5 differently.
Thanks, we have altered this in line with the previous comment.

P1 L30:*"These offsets suggest that, at least at lower water contents, $CO_2$-$H_2O$ isotope equilibration primarily occurs with water pools that are bound to particle surfaces, which are expected to be depleted in $^{18}O$ compared to bulk soil water."*

P4 L8: I am missing a clear research question here. You present the research gap, but do not state any hypothesis or research question before getting to the objectives in L9ff.
The main research question is whether soil CA activity can reasonably estimated from gas flux measurements in the absence of independent information about soil water isotopic composition. Secondly, we aim to better understand $CO_2$-$H_2O$ isotope equilibration in soil. This has been clarified in the text.

P4 L9-L16:*"Given the need to make an assumption about the soil water pool with which $CO_2$ is interacting, the potential for spatial and temporal variability of $\delta_{sw}$, and limited a priori information with respect to appropriate sampling resolution and depth (Miller et al., 1999; Riley, 2005), approaches allowing CA activity to be estimated in the absence of this information are desirable.*

*Here we test whether soil CA activity can be reasonably estimated in the absence of independent information about $\delta_{sw}$ and investigate assumptions about soil $CO_2$-$H_2O$ isotope equilibration. To do so we develop a novel approach to obtain solutions for $v_{inv}$ and $\delta_{eq}$, as a function of the response of $\delta_R$ to variations in $\delta_a$, from gas flux measurements. "*

P12 L16: I suggest providing statistical tests rather than using "broadly". Also for the L19 "distinct".
We have not conducted statistical tests on the slopes and intercepts as these could potentially vary as function of soil properties between incubations e.g. soil depth, bulk density etc. After taking these variances into account we test differences for our terms of interest (Table 3). The words 'broadly' and 'distinct' were chosen to clearly indicate that these visual descriptions of Figure 4 rather than statistical statements.

P12 L31: You do not present $\delta_{sm,eq}$ in the Figure 3. Please add.
We have not plotted the estimates of $\delta_{sw,eq}$ in Figure 3, as these reflect an integrated signal with depth and also make the plot harder to read (see Figure 3 MS below). Estimates of $\delta_{sw,eq}$ are provided in Table 3.

[Figure]

*"Figure 3 MS: Depth profiles of the $\delta^{18}O$ of soil water ($\delta_{sw}$). Points and error bars indicate mean and standard*

*deviation δ$_{sw}$ determined following cryogenic extraction of water (δ$_{sw,ce}$) from incubated soils at intervals of 0-1, 1-2, 2-3, and 4-5 cm below the surface. Shaded areas indicate mean and standard deviation δ$_{sw}$ determined to be in equilibrium with CO$_2$ (δ$_{sw,eq}$) from gas flux measurements. Colours indicate the three different irrigation water δ$^{18}$O (δ$_{iw}$) treatments."*

P14 L14: I do not like "immobile" water pool and encourage to use a different term, as the soil water held at low pressure heads is less mobile, but not stagnant. However, I know that this is widely used and common nomenclature is missing. Maybe "less mobile" or "water at lower pressure heads"? Or instead "mobile and immobile" using "bulk soil water"?

We agree that the terminology "mobile" and "immobile" water is too strong and misleading. We replaced it here with the terms macro-pore and micro-pore to better reflect the differences between relatively free and bound water pools.

P14 L30: *"Differences in the water pools characterised by different methodologies for determining the isotopic composition of soil waters are well known, with the cryogenic extraction method being expected to remove macro-pore, micro-pore, hygroscopic and potentially crystalline water, whilst the static equilibration of soils with CO$_2$ is expected to principally reflect only the macro-pore and micro-pore pools (Hsieh et al., 1998b; Orlowski et al., 2016b; Sprenger et al., 2015)."*

P15 L2-L6: This reads more like results and introducing a new figure would also better fit to the results section.

Classically, this is true. However, as these measurements were conducted post-hoc to test the explanation proposed in the discussion, we feel that the current placement better reflects the development of the work.

P15 L7: You do not have a data point at 95% water filled pore space. Therefore, I prefer you refer here to 75%.

We have amended the figure to the range of data shown (see below) and altered the text accordingly.

P15 L30: *" The fact that these relationships indicate the offset decreases at higher water contents may indeed support the inference that estimates of δ$_{sw,eq}$ are being influenced by fractionation between surface and bulk water pools."*

[Figure]

Table 2: Be consistent with the decimal places for the delta-values.
Done, thanks.

Table 3: In the 5$^{th}$ column, it should be "ã" not "Ã".
Done, thanks.

Figure 2: Is the dotted grey line showing the measurements at 1Hz and the dots, diamonds and triangles are showing the average values integrated over time?

The symbols indicate corrected average values. The uncorrected 1 Hz data is not plotted as it is difficult to coherently combine corrected and uncorrected values on the same plot. The dashed line is provided as a visual aid for sequence

order. We have amended the caption to clarify this point.

*"Figure 2: An example of the gas exchange measurement sequence, scanning sequentially calibration cylinders, the chamber line during a stabilisation period, calibration cylinders again, and finally the chamber and bypass lines, for the three different $\delta^{18}O$ of $CO_2$ delivered to the inlet of the incubation system ($\delta_b$). In this case, the $\delta_b$ inlet conditions, whose changes are indicated by the vertical dashed lines, started with $\delta_{b,med}$ and ended with $\delta_{b,low}$. Symbols represent the calibrated average values and the dotted line is provided as a visual aid and does not correspond to raw 1-Hz data, (a) total $CO_2$ concentration and, (b) $\delta^{18}O$ of $CO_2$."*

Figure 3: Consider adding the $\delta_{sm,eq}$ as you refer to that in the manuscript.
See above (P12 L31 comment)

---

## Author Comment (AC2) · 3 Oct 2017

We would like to thank the reviewer for their time in reviewing this manuscript. Please see the uploaded supplementary material for our responses to their comments.
* * *
[Figure]

Response to Anonymous Referee #2

We would like to thank Referee #2 for taking the time to review the manuscript. We have reproduced their comments, in blue, along with our responses in below.

To start with, the statistical analysis of the data should be improved, and that part of the Methods description should be elaborated and improved. As statistical method to assess the treatment effects in this study I recommend linear mixed effects models, see e.g. [Gueorguieva and Krystal, 2004; Crawley, 2009]. Crawley, M. J. (2009), The R book, 942 pp., John Wiley & Sons Ltd, Chichester. Gueorguieva, R., and J. H. Krystal (2004), Move over ANOVA, progress in analyzing repeated-measures data and its reflection in papers published in the archives of general psychiatry., Archives of General Psychiatry, 61, 310-317. Anon: Wiley: The R Book, 2nd Edition - Michael J. Crawley, [online] Available from: http://www.wiley.com/WileyCDA/WileyTitle/productCd-0470973927.html (Accessed 14 September 2017), n.d.
Thanks, we agree that a proper description of the statistics used was lacking and now have added a full description of our approach to the method section as suggested. We are not sure a mixed effect modelling approach is the best way forward for our data. We conducted a total of 18 incubations, with 6 incubations for each of the three levels (i.e. addition of $\delta_{sw-low}$, $\delta_{sw-med}$ and $\delta_{sw-high}$ water) of water treatment. Whilst repeated measurements were made (i.e. the gas fluxes at different inlet conditions) on each incubation these are reduced to single parameters when regression coefficients are calculated. We test whether there are significant differences between soil properties or model parameters (determined from these coefficients) among water treatments. As such, we consider the 18 incubations to be independent for these tests. For this reason and as we are testing for differences between three population means (of the same factor / categorical independent variable i.e. $\delta_{sw}$ treatment), we used one-way analysis of variance. We chose not report statistical test of treatment effects for the gas flux data shown in Table 2 (and section 3.3), however, the reviewer is correct that a mixed effect modeling approach would be appropriate here. Hopefully the suggested improvements to the methods clarify this point.

P10 L23: *"Treatment summaries are reported as mean and standard deviation unless stated otherwise. A total of 18 incubations were conducted on sub-samples of same homogenised bulk soil. Six independently replicated incubations were conducted for each of the three $\delta_{sw}$ water treatments. Soil properties and model parameters were determined individually for each incubation as described above. Differences in soil properties and model parameters among $\delta_{sw}$ treatments, with statistical significance reported at p < 0.01, were tested through one-way analysis of variance with post-hoc comparison by Tukey's HSD (Crawley, 2007; Mendiburu, 2016). To do so, a given property or parameter was taken as the dependent variable and $\delta_{sw}$ treatment as the categorical independent variable."*

I noted that the reference that is currently used in the Statistics part is missing on the reference list (Mendiburu, 2016).
The reference for Mendiburu was present but the new-line after the previous reference (Massman, 1998) was missing making it hard to see. We have corrected this, thanks.

Moreover, the Results section should be improved. In long parts many values are listed, e.g. means and error estimates for several parameters and treatments are spelled out in the text. I suggest to check which values are already given in the Tables, and to consider moving more of the values currently given in the text into Tables to refer to.
Following this advice we have removed duplicated numbers from the text and expanded Table 1.

Also, the authors are using many acronyms throughout the text. I find they are too many and this makes the text in parts hard to read. I suggest to reconsider which acronyms are central and to keep these, but consider to spell out certain variables (i.e. avoid too many acronyms). Alternatively, you might add a list of acronyms to the manuscript and refer to it repeatedly, to facilitate for the reader to look up the meaning of all acronyms during reading.
We agree that the manuscript makes use of several symbols that may need to be re-defined regularly to help the reader and, at the same time, we feel that the symbols used are vital to clearly relate to the methods without lengthening the text. For this reason we were careful to select consistent and logical symbols e.g. $\delta_{sw,cz}$ for soil water isotope composition determined following cryogenic extraction or $\delta_{sw,eq}$ for soil water isotope composition determined to be in equilibrium with $CO_2$ from gas flux measurements. However, we understand that following multiple symbols through a text can be difficult for the reader. In acknowledgement of this point, we have removed a number of less central symbols (e.g. $\delta_{kin}$, $k_{int/uncat}$ PTFE, GWC) and refer back to the meaning of important symbols at key points in the hope that this prevents the reader from having to search back through the text for first usage.

Please check as well that all acronyms are actually defined upon first use, and consider to even define acronyms that are common in your field but may not be obvious to all readers of the article (e.g. VPDBg and VSMOW-SLAP).
Done, thanks.

The same applies to the Tables and Figures, please include in footnotes or legend the meaning of the used acronyms (if you decide to keep them), with the goal that Figures and Tables can be understood independent of the text. As example I refer to the legend of Fig. 6, which contains four acronyms and is difficult to understand in its current form.

**Fig. 1.**

---

## Author Comment (AC3) · 3 Oct 2017

Response to Anonymous Referee #2

We would like to thank Referee #2 for taking the time to review the manuscript. We have reproduced their comments, in blue, along with our responses in below.

To start with, the statistical analysis of the data should be improved, and that part of the Methods description should be elaborated and improved. As statistical method to assess the treatment effects in this study I recommend linear mixed effects models, see e.g. [Gueorguieva and Krystal, 2004; Crawley, 2009]. Crawley, M. J. (2009), The R book, 942 pp., John Wiley & Sons Ltd, Chichester. Gueorguieva, R., and J. H. Krystal (2004), Move over ANOVA, progress in analyzing repeated-measures data and its reflection in papers published in the archives of general psychiatry., Archives of General Psychiatry, 61, 310-317.Anon: Wiley: The R Book, 2nd Edition - Michael J. Crawley, [online] Available from: http://www.wiley.com/WileyCDA/WileyTitle/productCd-0470973927.html (Accessed 14 September 2017), n.d.

Thanks, we agree that a proper description of the statistics used was lacking and now have added a full description of our approach to the method section as suggested. We are not sure a mixed effect modelling approach is the best way forward for our data. We conducted a total of 18 incubations, with 6 incubations for each of the three levels (i.e. addition of $\delta_{iw-low}$, $\delta_{iw-med}$ or $\delta_{iw-high}$ water) of water treatment. Whilst repeated measurements were made (i.e. the gas fluxes at different inlet conditions) on each incubation these are reduced to single parameters when regression coefficients are calculated. We test whether there are significant differences between soil properties or model parameters (determined from these coefficients) among water treatments. As such, we consider the 18 incubations to be independent for these tests. For this reason and as we are testing for differences between three population means (of the same factor / categorical independent variable i.e. $\delta_{iw}$ treatment), we used one-way analysis of variance. We chose not report statistical test of treatment effects for the gas flux data shown in Table 2 (and section 3.3), however, the reviewer is correct that a mixed effect modeling approach would be appropriate here. Hopefully the suggested improvements to the methods clarify this point.

P10 L23: *"Treatment summaries are reported as mean and standard deviation unless stated otherwise. A total of 18 incubations were conducted on sub-samples of same homogenised bulk soil. Six independently replicated incubations were conducted for each of the three $\delta_{iw}$ water treatments. Soil properties and model parameters were determined individually for each incubation as described above. Differences in soil properties and model parameters among $\delta_{iw}$ treatments, with statistical significance reported at p < 0.01, were tested through one-way analysis of variance with post-hoc comparison by Tukey's HSD (Crawley, 2007; Mendiburu, 2016). To do so, a given property or parameter was taken as the dependent variable and $\delta_{iw}$ treatment as the categorical independent variable."*

I noted that the reference that is currently used in the Statistics part is missing on the reference list (Mendiburu, 2016).

The reference for Mendiburu was present but the new-line after the previous reference (Massman, 1998) was missing making it hard to see. We have corrected this, thanks.

Moreover, the Results section should be improved. In long parts many values are listed, e.g. means and error estimates for several parameters and treatments are spelled out in the text. I suggest to check which values are already given in the Tables, and to consider moving more of the values currently given in the text into Tables to refer to.

Following this advice we have removed duplicated numbers from the text and expanded Table 1.

Also, the authors are using many acronyms throughout the text. I find they are too many and this makes the text in parts hard to read. I suggest to reconsider which acronyms are central and to keep these, but consider to spell out certain variables (i.e. avoid too many acronyms). Alternatively, you might add a list of acronyms to the manuscript and refer to it repeatedly, to facilitate for the reader to look up the meaning of all acronyms during reading.

We agree that the manuscript makes use of several symbols that may need to be re-defined regularly to help the reader and, at the same time, we feel that the symbols used are vital to clearly relate to the methods without lengthening the text. For this reason we were careful to select consistent and logical symbols e.g. $\delta_{sw,ce}$ for soil water isotope composition determined following cryogenic extraction or $\delta_{sw,eq}$ for soil water isotope composition determined to be in equilibrium with $CO_2$ from gas flux measurements. However, we understand that following multiple symbols through a text can be difficult for the reader. In acknowledgement of this point, we have removed a number of less central symbols (e.g. $\delta_{atm}$, $k_{iso,uncat}$, PTFE, GWC) and refer back to the meaning of important symbols at key points in the hope that this prevents the reader from having to search back through the text for first usage.

Please check as well that all acronyms are actually defined upon first use, and consider to even define acronyms that are common in your field but may not be obvious to all readers of the article (e.g. VPDBg and VSMOW-SLAP).

Done, thanks.

The same applies to the Tables and Figures, please include in footnotes or legend the meaning of the used acronyms (if you decide to keep them), with the goal that Figures and Tables can be understood independent of the text. As example I refer to the legend of Fig. 6, which contains four acronyms and is difficult to understand in its current form.

Following this good advice we have updated Table and Figure captions accordingly.

"Table 1: Soil properties by irrigation water ($\delta_{iw}$) treatment . Means (n = 6) and standard deviations (in parentheses) for maximum soil depth ($z_{max}$), total porosity ($f_t$), and volumetric soil water content ($q_w$). Lower-case letters indicate significant differences (one-way analysis of variance and Tukey's HSD,  p < 0.01) among $\delta_{iw}$ treatments."

"Table 2: Gas exchange data by irrigation water ($\delta_{iw}$)  treatment at the three different incubation system inlet $CO_2$ ($\delta_b$) conditions. Means and standard deviations (in parenthesis) for total $CO_2$ concentration in the bypass ($C_b$) and the chamber ($C_a$), the $\delta^{18}O$ of $CO_2$ in the bypass ($\delta_b$) and the chamber ($\delta_a$) and, the net flux of $CO_2$ ($F_R$) and its $\delta^{18}O$ signature ($\delta_R$)."

"Table 3: Model solutions by irrigation water ($\delta_{iw}$) treatment. Means (n = 6) and standard deviations (in parenthesis) for the piston velocity of $CO_2$ assuming a semi-infinite soil depth ($v_{inv}$), the piston velocity of $CO_2$ assuming a finite soil depth ($\tilde{v}_{inv}$), the apparent rate of $^{18}O$ exchange between $CO_2$ and soil water ($k_{iso}$), the effective diffusional fraction of $CO_2$ assuming a finite soil depth ($\tilde{a}$), and the $\delta^{18}O$ of soil water in equilibrium with $CO_2$ as determined from gas flux measurements ($\delta_{sw,eq}$). Lower-case letters indicate significant differences (one-way analysis of variance and Tukey's HSD,  p < 0.01) among $\delta_{iw}$ treatments."

"Figure 1: Schematic of the system used to make gas exchange measurements.  Alternate measurements of the concentration and $\delta^{18}O$ of  $CO_2$ in chamber ($C_a$, $\delta_a$) and bypass lines ($C_b$, $\delta_b$) are made under inlet conditions that differ in terms of  the $\delta^{18}O$ of $CO_2$. "

"Figure 2: An example of the gas exchange measurement sequence, scanning sequentially calibration cylinders, the chamber line during a stabilisation period, calibration cylinders again, and finally the chamber  and bypass lines, for the three different $\delta^{18}O$ of $CO_2$ delivered to the inlet of the incubation system ($\gtrless_b$). In this case, the $\delta_b$ inlet conditions, whose changes are indicated by the vertical dashed lines, started with $\delta_{b,med}$ and ended with $\delta_{b,low}$. Symbols represent the calibrated average values and the dotted line is provided as a visual aid and does not correspond to raw 1-Hz data, (a) total $CO_2$ concentration and, (b)  $\delta^{18}O$ of $CO_2$."

"Figure 3: Incubation depth profiles of the $\delta^{18}O$ of cryogenically extracted soil water ($\delta_{sw,ce}$), at  intervals of 0-1, 1-2, 2-3, and 4-5 cm below the surface. Symbols and error bars indicate means and standard deviations by irrigation water ($\delta_{iw}$) treatment and depth interval."

"Figure 4: Relationships between the $\delta^{18}O$ of soil-atmosphere $CO_2$ exchange ($\delta_R$) and the $\delta^{18}O$ of $CO_2$ in the chamber line ($\delta_a$)  by irrigation water ( $\delta_{iw}$) treatment. Symbol shapes indicate measurements made at different inlet conditions ($\delta_b$) that varied in terms of their $\delta^{18}O$ of $CO_2$. Dashed lines indicate linear regressions for individual incubations."

"Figure 5: Relationships between water-filled pore space and the difference between estimates of the $\delta^{18}O$ of soil water in equilibrium with $CO_2$ as estimated from gas flux measurements ($\delta_{sw,eq}$) and that estimated by cryogenic extraction ($\delta_{sw,ce}$)  at depths of 0-1 cm (squares) and 1-5 cm (circles). Dashed lines and shaded areas indicate the linear regressions and associated 95 % confidence intervals for the two sampling depths."

"Figure 6: Model relationships between the apparent rate of $^{18}O$ exchange ($k_{iso}$) between $CO_2$ and soil water  and the $\delta^{18}O$ of soil water in equilibrium with $CO_2$ ($\delta_{eq,ce}$). These $\delta_{eq,ce}$ values were assumed from the depth averaged (0 to 5 cm) $\delta^{18}O$ of cryogenically extracted water for the incubations that received the $\delta_{iw,low}$ ($\delta^{18}O$ of -6.74 ± 0.03 ‰ VSMOW-SLAP) irrigation water treatment. Colours indicate the different responses for the same set of incubations at the three inlet conditions that differed by their  $\delta^{18}O$ composition of $CO_2$ ($\delta_b$)."

*In Figure 5, please add confidence intervals to the regression lines. This may not be feasible in terms of clarity for Fig. 4, which contains many regression lines in one graph. In that case please add a note to the legend of Fig. 4 why confidence intervals are not shown.*

We have not added these to Figure 4 for the reason indicated. We have added confidence intervals to Figure 5 and updated the caption accordingly.

[Figure]

*"Figure 5: Relationships between water-filled pore space and the difference between estimates of the $\delta^{18}O$ of soil water in equilibrium with $CO_2$ as estimated from gas flux measurements ($\delta_{sw,eq}$) and that estimated by cryogenic extraction ($\delta_{sw,ce}$) at depths of 0-1 cm (squares) and 1-5 cm (circles). Dashed lines and shaded areas indicate the linear regressions and associated 95 % confidence intervals for the two sampling depths."*

Please add a reference for the assumed particle density [Linn and Doran, 1984]. Linn, D. M., and J.W. Doran (1984), Effect of water-filled pore space on carbon dioxide and nitrous oxide production in tilled and nontilled soils., Soil Science Society of America Journal, 48, 1267-1272
Done, thanks.

*"Total porosity ($\phi_t$) was calculated from bulk density assuming a particle density of 2.65 g cm$^{-3}$ (Linn and Doran, 1984)."*

*"Linn, D. M. and Doran, J. W.: Effect of Water-Filled Pore Space on Carbon Dioxide and Nitrous Oxide Production in*

*Tilled and Nontilled Soils, Soil Science Society of America Journal, 48(6), 1267–1272,*

*doi:10.2136/sssaj1984.03615995004800060013x, 1984."*

P1/L11: Move the comma: "..., a group of enzymes that catalyse the hydration of CO2 in soils and plants,..."
Done, thanks.

*"To do so, the activity of carbonic anhydrases (CA), a group of enzymes that catalyse the hydration of $CO_2$ in soils and plants, needs to be understood."*

P5/20: "were monitored" (change from "was")
Done, thanks.

*"Relative humidity and temperature inside the humidifier were monitored using a small combined sensor and data-logger (Hydrochron, iButtonLink, LLC., USA)."*

P9/L13: "R Development Core Team"
The citation provided by the citation function in R or indicated by the R-project website ([https://cran.r-project.org/doc/FAQ/R-FAQ.html#Citing-R](https://cran.r-project.org/doc/FAQ/R-FAQ.html#Citing-R)) uses 'R Core Team' rather than 'R Development Core Team'. We have added the relevant version information.

*"All data processing and analysis was conducted in R version 3.3 (R Core Team, 2017)."*